# Tuning reactivity of Fischer–Tropsch synthesis by regulating TiO$_x$ overlayer over Ru/TiO$_2$ nanocatalysts

Yaru Zhang[1,2], Xiaoli Yang[1,2,3], Xiaofeng Yang [1✉], Hongmin Duan[1], Haifeng Qi[1,2], Yang Su[1], Binglian Liang[1], Huabing Tao[4], Bin Liu[4], De Chen[3], Xiong Su [1✉], Yanqiang Huang [1✉] & Tao Zhang[1]

The activity of Fischer–Tropsch synthesis (FTS) on metal-based nanocatalysts can be greatly promoted by the support of reducible oxides, while the role of support remains elusive. Herein, by varying the reduction condition to regulate the TiO$_x$ overlayer on Ru nanocatalysts, the reactivity of Ru/TiO$_2$ nanocatalysts can be differentially modulated. The activity in FTS shows a volcano-like trend with increasing reduction temperature from 200 to 600 °C. Such a variation of activity is characterized to be related to the activation of CO on the TiO$_x$ overlayer at Ru/TiO$_2$ interfaces. Further theoretical calculations suggest that the formation of reduced TiO$_x$ occurs facilely on the Ru surface, and it involves in the catalytic mechanism of FTS to facilitate the CO bond cleavage kinetically. This study provides a deep insight on the mechanism of TiO$_x$ overlayer in FTS, and offers an effective approach to tuning catalytic reactivity of metal nanocatalysts on reducible oxides.

[1] State Key Laboratory of Catalysis, Dalian Institute of Chemical Physics, Chinese Academy of Sciences, Dalian 116023, China. [2] University of Chinese Academy of Sciences, Beijing 100049, China. [3] Department of Chemical Engineering, Norwegian University of Science and Technology, Trondheim 7494, Norway. [4] School of Chemical and Biomedical Engineering, Nanyang Technological University Singapore, 637459 Singapore, Singapore. ✉email: yangxf2003@dicp.ac.cn; suxiong@dicp.ac.cn; yqhuang@dicp.ac.cn

Fischer–Tropsch synthesis (FTS) offers a powerful way to convert syngas (a mixture of CO and $H_2$) to long-chain hydrocarbons, by which the transformation of nonpetroleum resources (derived from coal, natural gas, or biomass) into high value-added chemicals and fuels becomes economically efficient and environmentally friendly[1–3]. The most challenging step in this process is consented as the CO activation, with which the further hydrogenation to $CH_x$ ($x = 1, 2, 3$) species for the carbon chain growth becomes facile on the surfaces of late transition metals like Fe, Co, Ru, and Rh[4–7]. Among these metals, Ru is identified to be intrinsic of high activity and selectivity in FTS[8]. In particular, large particle sizes of Ru (~8 nm) are highly desirable[9–11], on which the direct or H-assisted CO dissociation is greatly facilitated[12,13]. As such, great efforts have been made on modulating the surface structure of Ru-based catalysts, including by tuning particle size[9–11], changing crystal phase[14], and varying exposed plane[15], to further promote the catalytic performance of Ru catalysts.

Beyond that, there are strong metal–support interactions (SMSI) on supported metal catalysts when reducible oxide such as $TiO_2$ is used as a support[16,17]. More specifically, reducible oxide migrates to the metal surface by forming a thin overlayer under the reduction condition[18], which then results in a unique metal/support interface and variegates the behavior of catalyst in reactions[19–21]. The utilization of such SMSI has also been demonstrated to be an alternative strategy to enhance the catalytic reactivity of metal catalysts in FTS[22,23]. This promoted effect is generally attributed to the interface between metal and support, which serves as a new active site with an improved activity in the reactions[18,23]. However, the catalytic mechanism and the intrinsic roles of such newly generated interfaces are remaining elusive.

In this research, by regulating a $TiO_x$ overlayer on the Ru nanoparticles (~2 nm), the catalytic activity of this Ru/$TiO_2$ nanocatalyst in FTS can be boosted. With various characterizations and theoretical modeling, the reduced $TiO_x$ overlayer on Ru nanocatalysts is demonstrated to participate in and dramatically facilitate the bond cleavage of CO. The results of this work are expected to play an important role in the mechanism understanding of SMSI in FTS, and also provide guidance with regard to tuning the catalytic properties of metal nanocatalysts supported on reducible oxides.

## Results

**Structural characterization.** With the aim to investigate the effect of $TiO_x$ overlayer covered over the Ru nanoparticles (NPs) on the reactivity in FTS, in the present work, we use wet impregnation method to fabricate Ru-based catalysts with small-sized metal nanoparticles by using rutile $TiO_2$ as a support[24]. In this text, catalysts reduced at specific temperatures were denoted as Ru/$TiO_2$-$x$, where $x$ refers to reduction temperatures ($x = 200$–600). The loading of Ru, after calcination in air and the following thorough chlorides removal process, has been determined to be 2.2 wt% by ICP-OES. As indicated by Brunauer–Emmett–Teller (BET) testing (Supplementary Table 1), different Ru/$TiO_2$-$x$ samples are similar in physical textures, with almost identical surface areas and pore volumes.

Meanwhile, all Ru/$TiO_2$ samples are found to possess analogous morphologies of metal NPs, that is, the Ru NPs are all well dispersed on the support and have a uniform size distribution with an average diameter of ~2 nm (Fig. 1a, b and Supplementary Figs. 1–3). It was attributed to the usage of rutile $TiO_2$ as the support in our research, which possesses the same crystal phase and comparable lattice parameters with that of $RuO_2$ (Supplementary Table 2). As a result, the calcination step by formation of $RuO_2$/$TiO_2$ interphase helps to resist the aggregation of metal nanoparticles even after the high-temperature pretreatment (Supplementary Fig. 4). On the other hand, however, there are significant discrepancies on the micro-structures of Ru NPs on the $TiO_2$ support when pretreated at different reduction temperatures (Fig. 1c and Supplementary Fig. 5). In detail, a distinct morphology of Ru NP can be resolved on the support when sample was reduced at a temperature below 300 °C, while a visible coating on the Ru NPs can be distinguishable after higher temperature pre-reduction. In terms of the SMSI between Ru and rutile $TiO_2$, it was ascribed to the $TiO_x$ overlayer over the Ru NPs under high-temperature reduction condition, and the migration of $TiO_x$ over Ru NPs was initiated at a reduction temperature higher than 300 °C.

To get a qualitative comparison of the exposure of Ru species after coating by $TiO_x$ overlayer, the chemisorption of CO and $H_2$ were measured to estimate the Ru dispersion on these Ru/$TiO_2$-$x$ samples. As seen in Table 1, the values obtained by different probe molecules give the same tendency of the metal dispersions

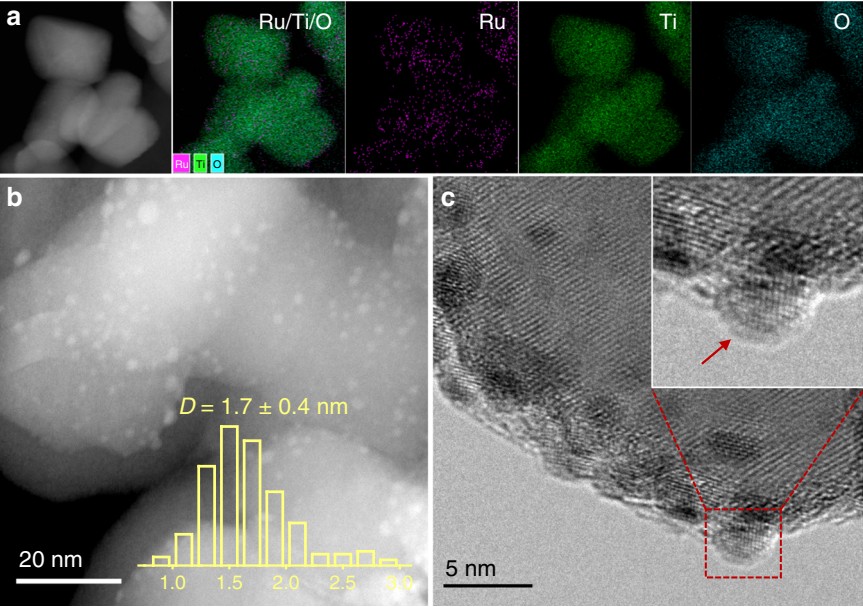

**Fig. 1 Morphological study of the Ru/TiO₂ catalysts. a** Elemental mapping of Ru/Ti/O in the fresh Ru/TiO₂ catalyst. **b** HAADF-STEM image of the Ru/TiO₂ catalyst pretreated at 600 °C (Ru/TiO₂-600 sample) with the metal size distribution. **c** HRTEM image of the Ru/TiO₂-600 catalyst.

for different Ru/TiO$_2$-$x$ samples, that is, the dispersion of Ru decreases with increasing the reduction temperature from 300 to 600 °C. This can be explained by a gradual encapsulation of the Ru NPs by TiO$_x$ overlayer as increasing the reduction temperature from 300 to 600 °C, which was in good agreement with the TEM observations. As compared, the dispersion derived from H$_2$ chemisorption was lower than that from CO chemisorption. It might be caused by the Ru$^{n+}$ sites at the Ru–TiO$_2$ interface, which are unavailable for H$_2$ chemisorption due to the SMSI effects[25], but it can be readily involved in CO chemisorption as indicated in our further in situ DRIFT spectra experiments. Even so, the H$_2$ uptakes on the Ru/TiO$_2$-200 sample was found to be less than that of the Ru/TiO$_2$-300. This can be explained by the results of H$_2$ temperature-programmed reduction (H$_2$-TPR), in which the predominant reduction of Ru/TiO$_2$ to metallic Ru occurring at a temperature higher than 200 °C (Supplementary Fig. 6). In addition, the decline of the surface metallic Ru exposure by a gradual encapsulation of the Ru NPs by TiO$_x$ overlayer as increasing the reduction temperature was also confirmed by the underpotential deposition of copper (Cu upd) experiments, with which the metallic surface area can be semi-quantified in terms of the integral area of current for the reduction deposition of copper on the exposed metal surface[26] (Supplementary Fig. 7 and Supplementary Table 3).

The evolution of Ru/TiO$_2$ catalysts at different temperature reductions was also investigated with X-ray absorption spectroscopy (XAS). The extended X-ray absorption fine structure (EXAFS) of Ru $K$-edge and the fitting results (Fig. 2a and Supplementary Table 4) have demonstrated that the coordination number (CN) associated with Ru–Ru pair (~2.67 Å) is increased

gradually from 2.2 to 5.3 as the reduction temperature increased from 200 to 600 °C, while the CN of the Ru–O pair (~1.98 Å) presents an inverse tendency by decreasing from 4.0 to 2.4. It suggests a gradual improvement of the reduction degree of ruthenium oxide to metallic phase. Correspondingly, a shift of edge energy towards Ru foil was observed by the X-ray absorption near-edge structure (XANES) (Supplementary Fig. 8). Nevertheless, the inevitable Ru–O bonding for Ru/TiO$_2$ samples indicates a strong interfacial interaction between Ru and TiO$_2$, further confirming the formation of TiO$_x$ coating on the Ru NPs. On the other hand, the soft XANES spectra at the Ti $L_{3,2}$-edge (Fig. 2b) exhibit a decline in peak intensities with the increase of pretreatment temperature, indicative of an increasing degree of the reduction of TiO$_2$, owing to the formation of TiO$_x$ overlayer on Ru NPs. The growth of the reduced TiO$_x$ overlayer was also evidenced by the increased concentration of Ti$^{3+}$ species accompanying by the decreased ratio of surface Ru/Ti estimated from the Ti-2$p$ and Ru-3$p$ in XPS results (Supplementary Fig. 9 and Supplementary Table 5).

On the basis of the above observations, the structure evolution of Ru/TiO$_2$ at different stages of reduction was then suggested in Fig. 2c. Benefiting from the lattice matching of RuO$_2$/TiO$_2$ interphase, a small size of Ru NPs with the improved sintering resistance can be facilely acquired by the following reduction pretreatment. In the initial step of reduction, e.g., Ru/TiO$_2$-300, a dominant metallic Ru will be exposed, and it serves as a typical Ru-based nanocatalysts in FTS. With further increasing the reduction temperature, the SMSI between Ru and TiO$_2$ governs the surface exposure of Ru NPs, and the TiO$_x$ thin layer begins to migrate and coating the Ru surface, resulting in a shrinkage of

**Table 1 H$_2$ and CO chemisorption results for different Ru/TiO$_{2-x}$ samples.**

| Sample | CO uptake ($\mu mol_{CO}\ g_{Ru}^{-1}$) | Ru dispersion by CO uptake | H$_2$ uptake ($\mu mol_{H2}\ g_{Ru}^{-1}$) | Ru dispersion by H$_2$ uptake |
|---|---|---|---|---|
| Ru/TiO$_2$-200 | 102.6 | 47.2% | 28.1 | 25.8% |
| Ru/TiO$_2$-300 | 94.2 | 43.3% | 32.2 | 29.6% |
| Ru/TiO$_2$-400 | 84.7 | 38.9% | 27.5 | 25.3% |
| Ru/TiO$_2$-450 | 74.0 | 34.0% | 21.6 | 19.9% |
| Ru/TiO$_2$-500 | 59.8 | 27.5% | 17.4 | 16.0% |
| Ru/TiO$_2$-600 | 38.0 | 17.5% | 10.8 | 9.9% |

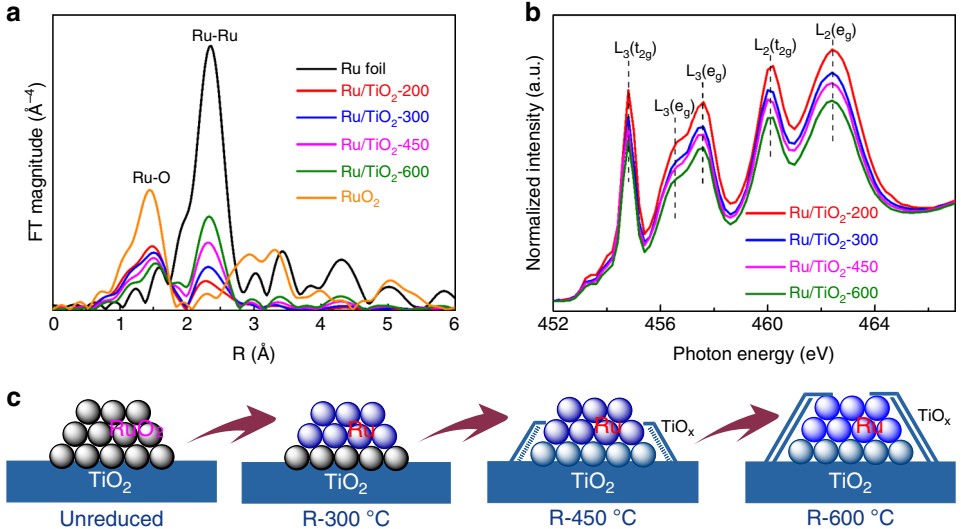

**Fig. 2 The evolution of Ru/TiO$_2$ catalysts reduced at different temperatures. a** Fourier transforms of the $k^3$-weighted EXAFS of Ru $K$-edge for Ru foil, RuO$_2$, and the Ru/TiO$_2$ catalysts pretreated at different temperatures (Ru/TiO$_2$-$x$ samples). **b** Ti $L_{3,2}$-edge XANES for the Ru/TiO$_2$ catalysts pretreated at different temperatures (Ru/TiO$_2$-$x$ samples). **c** A schematic illustration of the structural evolution of Ru/TiO$_2$ at different stages of reduction.

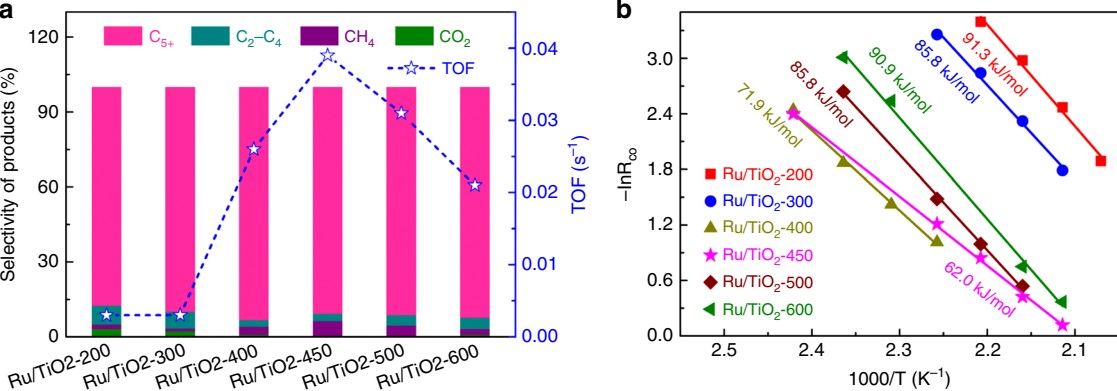

**Fig. 3 Catalytic results of the Ru/TiO₂ catalysts for Fischer–Tropsch synthesis. a** Catalytic performance of the Ru/TiO₂ catalysts pretreated at different temperatures (Ru/TiO₂-$x$ samples). Reaction conditions: 2 MPa, 160 °C, space velocity = 1200–6000 mL h⁻¹ g$_{cat}$⁻¹, H₂/CO/Ar = 64/32/4. **b** Arrhenius plots for CO hydrogenation over the Ru/TiO₂-$x$ catalysts.

metallic Ru surface by the TiO$_x$ overlayer (like that on Ru/TiO₂-450). Finally, for the Ru/TiO₂-600 sample, an excessive coverage of TiO$_x$ on Ru NPs causes a dominant TiO$_x$ overlayer on Ru nanocatalysts. Accordingly, a tunable degree of TiO$_x$ overlayer on Ru NPs can be easily achieved by varying the pretreatment condition, and it thus provides us an opportunity to explore the effect of metal/support interface of Ru/TiO₂ on the activity in FTS.

**Catalytic performance.** The catalytic performances of various Ru/TiO₂-$x$ catalysts in FTS were then evaluated at 160 °C with a 2 MPa reaction pressure according to our preliminary optimization of experiment conditions for achieving a high C₅₊ selectivity under a relatively mild condition (Supplementary Fig. 10). Notably, as shown in Fig. 3a and Supplementary Figs. 11, 12, all Ru/TiO₂-$x$ catalysts possess an excellent C₅₊ selectivity with a value up to 90%, indicating the promising application prospect of Ru/TiO₂ in FTS for high-carbon products. While the intrinsic reaction rate (reflected as the TOF value) greatly relies on the pretreatment temperature, and shows a volcano-like trend with the increase of reduction temperature (Fig. 3a, Supplementary Fig. 13 and Supplementary Table 6). As compared, samples of Ru/TiO₂-200 and Ru/TiO₂-300 manifest a much lower intrinsic activity (0.003 s⁻¹). A great promotion of activity was observed for the Ru/TiO₂-400 catalyst. Among these catalysts, the Ru/TiO₂-450 exhibits the highest activity with an intrinsic TOF value of 0.039 s⁻¹, which was also superior to other Ru-based catalysts reported previously (Supplementary Table 7). The further increasing of pretreatment temperature, however, causes an activity decay of catalysts, with a TOF value of only 0.021 s⁻¹ for Ru/TiO₂-600. Correspondingly, a reverse variation of apparent activation energy ($E_a$) was obtained, i.e., the Ru/TiO₂-450 presents the lowest $E_a$ with a calculated value of 62.0 kJ mol⁻¹, which was much lower than the values of other Ru/TiO₂ catalysts (Fig. 3b).

The particle size of metal was essential in determining the performance of Ru-based catalysts in FTS. The activity increases as the particle size of Ru nanocatalyst increased, with the small-sized Ru NPs behaving a rather poor activity[9–11]. This explains well the low activity of our small-sized Ru/TiO₂ catalysts reduced at low temperatures (Ru/TiO₂-200 and Ru/TiO₂-300 samples) as well as Ru NPs supported on the irreducible support (Ru/Al₂O₃-450 in Supplementary Fig. 14 and Supplementary Table 8). While an enhanced activity of Ru/TiO₂-450 catalyst suggests that the TiO$_x$ overlayer on Ru NPs has a positive contribution on the reactivity of Ru nanocatalysts. However, in terms of the decline of

activity for the Ru/TiO₂-600 catalyst covered dominantly by TiO$_x$ overlayer, the only TiO$_x$ overlayer cannot achieve a high activity for the FTS reaction. In this regard, the interface between metal and support of Ru/TiO₂ plays a crucial role on the activity promotion, where both the metallic Ru and TiO$_x$ overlayer are indispensable. The optimized composition of TiO$_x$ overlayer and Ru NPs on Ru/TiO₂-450 catalyst makes it possess an enhanced activity.

Furthermore, the Ru/TiO₂-450 catalyst also owns an excellent stability in the steady running state of FTS (Supplementary Figs. 10, 11). HAADF-STEM image of the spent Ru/TiO₂-450 catalyst suggests that the size of Ru can keep constant after testing (Supplementary Fig. 15). This was also benefited from the SMSI in the Ru/TiO₂-450 catalyst, which greatly prohibits the size aggregation of Ru during FTS process.

**Catalytic mechanism.** The role of the TiO$_x$ overlayer was then studied by the steady-state isotopic transient kinetic analysis (SSITKA)[27,28], by which the evolution of intermediates with the associated coverage and reactivity can be acquired (Supplementary Fig. 16). Limited by the atmospheric pressure condition in this analysis, as shown in Supplementary Fig. 17, CH₄ selectivity has an increase because of the preference of hydrogenation over C–C coupling for CH$_x$ intermediates. Notably, a good correlation between the intrinsic activity (TOF) of CO consumption and methane generation can be set up. As such, the coverage of CH$_x$ (represented as $\theta_{CH4}$ in SSITKA) was determined as a function of reduction temperature of Ru/TiO₂ (Fig. 4a). From our results, the activity improvement of Ru/TiO₂-450 can be attributed to the increased coverage of CH$_x$ intermediates on the catalyst surface[29]. By considering the comparable size of Ru for different Ru/TiO₂ samples, a promoted effect toward CO activation to generate CH$_x$ intermediates with the aid of TiO$_x$ overlayer can be expected on the Ru/TiO₂-450 catalyst.

To confirm our proposed mechanism of CO activation, the micro-calorimetry toward CO was measured for Ru/TiO₂ samples (Fig. 4b, c). The amount of CO chemisorption follows a trend of Ru/TiO₂-300 > Ru/TiO₂-450 > Ru/TiO₂-600. This can be explained by the decrease exposure of Ru for serving as the adsorbed sites toward CO as increasing reduction temperature. In particular, as compared with other catalysts, Ru/TiO₂-450 owns a large portion of CO chemisorption at a relative higher differential heat (>150 kJ mol⁻¹). This was attributed to the CO chemisorption on the interface site, followed by a dissociation of CO with the aid of TiO$_x$ overlayer. For comparison, the Ru/TiO₂-300 shows a predominant moderate chemisorption toward CO, with a

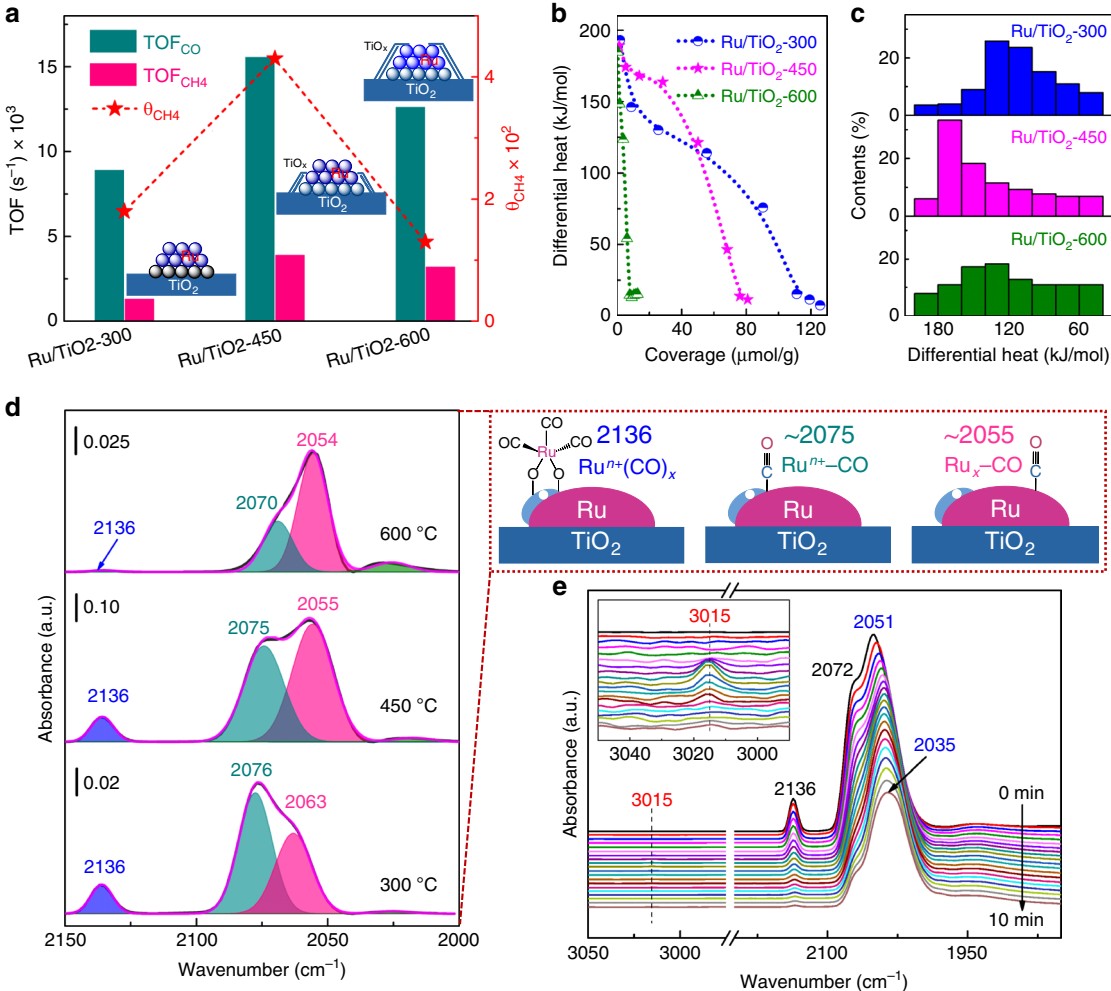

**Fig. 4 The participation of the TiOₓ overlayer in the C–O bond dissociation. a** The correlation between the intrinsic activity (TOF$_{CO}$ and TOF$_{CH4}$) and the coverage of active surface CH$_x$ intermediates (represented by $\theta_{CH4}$) as obtained by SSITKA experiments. Performing condition: 200 °C, 0.185 MPa, H$_2$/CO = 10. **b, c** Microcalorimetric measures of CO chemisorption and the distribution of differential heat on the Ru/TiO$_2$-x samples. **d** In situ DRIFT spectra obtained after CO adsorption and evacuation with helium at 160 °C, over the Ru/TiO$_2$-x catalysts. **e** Evolution of the CO$_{ad}$ species during H$_2$ flow at 160 °C as determined using in situ DRIFT spectra, over the Ru/TiO$_2$-450 catalyst.

differential heat of 120–150 kJ mol$^{-1}$ for CO chemisorption on the Ru sites. In contrast, the over-coverage of TiO$_x$ on Ru NPs causes a shortage of both the Ru sites and the interfaces for CO chemisorption/dissociation on Ru/TiO$_2$-600.

In situ diffuse reflectance infrared Fourier transform (DRIFT) spectra of CO chemisorption on Ru/TiO$_2$-x indicate that there are three distinct $\nu_{CO}$ bands located at approximately 2136, 2075, and 2056 cm$^{-1}$ in the carbonyl region (Fig. 4d). Here, the bands at 2136 and 2075 cm$^{-1}$ were often observed by performing CO adsorption on well-dispersed, partially oxidized Ru$^{n+}$ with a low coordination environment, which therefore were ascribed to multi-carbonyl (Ru$^{n+}$(CO)$_x$) and mono-carbonyl (Ru$^{n+}$–CO) species adsorbed on partially oxidized Ru$^{n+}$ sites on the interface, respectively[30–32]. While the peak at 2056 cm$^{-1}$ was assigned to linear CO adsorption on metallic Ru (Ru$_x$–CO)[33,34]. After purging H$_2$ into the CO-saturated Ru/TiO$_2$, the gaseous CH$_4$ product with a characterized frequency at 3015 cm$^{-1}$ was detected[35], accompanying by the consumption of CO (Fig. 4e and Supplementary Fig. 18). More importantly, by the complete consumption of CO of Ru$^{n+}$(CO)$_x$ and Ru$^{n+}$–CO, the further conversion of CO was restrained, with Ru$_x$–CO as a predominant chemisorption species on Ru surface. In this case, the interface of partially oxidized Ru$^{n+}$ sites were supposed to be the active sites

for the FTS reaction. As such, the intensity of CO related to FTS on the Ru/TiO$_2$-450 was found to be more remarkable than that of Ru/TiO$_2$-300 and the Ru/TiO$_2$-600 catalysts (Fig. 4d), which was responsible for its higher activity in FTS.

According to our results, a catalytic mechanism for CO transformation on the Ru/TiO$_2$-x catalysts was then proposed. Due to the SMSI over Ru/TiO$_2$, the TiO$_x$ overlayer on Ru NPs provides oxygen vacancies for anchoring the oxygen atoms from the dissociation of carbonyl group; it thus greatly facilitates the dissociation of CO on the Ru/TiO$_x$ interface of catalysts as also suggested by Bell and coworkers[36], by which the hydrogenation and C–C coupling can be realized on the Ru sites to produce carbon chain products. As for Ru/TiO$_2$-450, the optimized TiO$_x$ overlayer on Ru NP offers it a maximized activity in FTS, while the shortage of interface on both the Ru/TiO$_2$-300 and Ru/TiO$_2$-600 samples leads to a lower activity in FTS. As such, the participation of the TiO$_x$ overlayer in the C–O bond dissociation process was responsible for the superior reactivity of the Ru/TiO$_2$-450 catalyst.

**DFT calculations**. Theoretically, we have performed a density functional theory (DFT) study on the CO activation on the model catalyst on a TiO$_x$ cluster decorated Ru(001) surface. In Fig. 5a

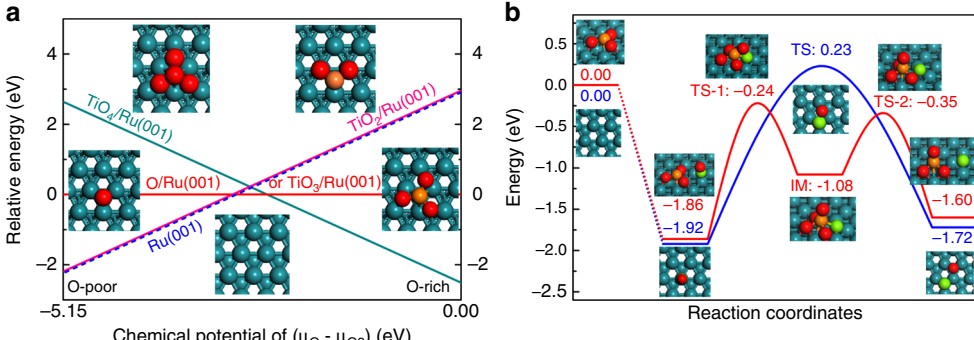

**Fig. 5 Theoretical study of the CO activation on a TiO$_x$ cluster decorated Ru(001) surface. a** Thermodynamic stability of different TiO$_x$/Ru(001) and O/Ru(001) under a variation of the chemical potential of O as referring to TiO$_3$/Ru(001) and Ru(001), respectively, with the atomic configuration in insets. Atom key: Ru (dark blue), O (red), Ti (orange), and C (green). **b** A possible catalytic mechanism of CO activation at GW level calculations on the TiO$_3$/Ru (001) model surface (red line), with the dissociation of CO on Ru(001) surface as a comparison (blue line).

and Supplementary Table 9, we have first studied the thermo-dynamic stability of different TiO$_x$ clusters on the Ru(001) surface at different oxygen chemical potential. As compared with the TiO$_6$ unit in bulk phase of rutile TiO$_2$, the TiO$_x$ was reduced facilely on the Ru(001) surface, and the TiO$_4$ cluster was found to be dominant under the oxygen-rich condition. By decreasing the oxygen chemical potential under reduction condition, the reduction of TiO$_4$ occurred readily on the Ru surface through a sequential reduction to TiO$_3$/Ru(001) and TiO$_2$/Ru(001), respectively. This was consistent with the experimental observation of a reduction of the TiO$_x$ overlayer under the reduction condition. On the TiO$_3$/Ru(001) surface, the activation of CO by C–O bond cleavage was then estimated, with that occurring on parent Ru(001) surface as a comparison. As we can see in Fig. 5b, the CO bond cleavage on the Ru (001) surface has a much high barrier (2.15 eV), and the laying down of the atop *CO adspecies adsorbed on the surface is the main obstacle during its dis-sociation, in good agreement with the previous results[37,38]. In contrast, with a reduced TiO$_3$ cluster decorating on the Ru(001) surface, the bond cleavage of CO adsorbed on the Ru site of interface can be greatly promoted by experiencing a calculated barrier of 1.62 eV, with the aid of TiO$_3$ as the O seizer of carbonyl group to transform to TiO$_4$. Taking into account that our experiments of FTS were conducted at a reaction temperature of 160–200 °C and a reaction pressure of 2 MPa, such a barrier is facile to overcome on the Ru/TiO$_2$ catalysts at the reaction con-dition of FTS. The as-dissociated C* adspecies can then be con-veniently diffused from the TiO$_x$/Ru(001) interface to Ru(001) (0.73 eV) for the further hydrogenation. Meanwhile, the reduc-tion of TiO$_4$/Ru(001) to TiO$_3$/Ru(001) was even thermo-dynamically more favorable than the surface reduction of O adspecies on the parent Ru(001) surface (Fig. 5a), which can then facilitate the catalytic cycle of CO activation on the interface. As indicated by the previous reports, such C* species is promising to be hydrogenated to CH$_x$ species and realize the C–C coupling on the Ru surface to produce the C$_{2+}$ products[39]. Accordingly, the Ru/TiO$_2$-450 catalyst, owing to its optimized Ru/TiO$_x$ interface for CO activation, shows a superior activity to other Ru/TiO$_2$-$x$ catalysts in FTS.

In conclusion, we have successfully fabricated a highly active Ru/TiO$_2$ catalyst for FTS by fine-tuning the catalyst pre-reduction condition. With increasing the reduction temperature, the TiO$_x$ overlayer is gradually enveloping the Ru NPs. The catalyst reduced at 450 °C exhibits a high intrinsic activity under mild conditions with a low apparent activation energy. The participa-tion of TiO$_x$ overlayer in promoting CO dissociation plays a vital role in activity enhancement during FTS. An optimized TiO$_x$

overlayer on the Ru NPs formed during reduction at 450 °C is evidently able to capture oxygen from carbonyl group adsorbed on the interface of Ru/TiO$_x$. This, in turn, facilitates the cleavage of C–O bonds. This work not only provides an understanding of the mechanism of CO activation on Ru/TiO$_2$ catalysts, but also suggests an effective approach to tailoring the catalytic properties of metal nanocatalysts supported on reducible oxides.

## Methods

**Catalyst preparation.** The Ru/TiO$_2$ catalysts were prepared by an impregnation method using rutile TiO$_2$ as the support. In a typical synthesis, 1.6 g of the aqueous RuCl$_3$·3H$_2$O solution (8.2 wt%, 0.0317 g Ru per gram of solution, AR) was diluted to 30 mL with deionized water. 2.0 g of rutile TiO$_2$ (40 nm, 99.8% metal basis) was then added to the solution and the resulting suspension was evaporated to dry in a 50 °C water bath with the vigorous stirring. The resulting solid was dried at 120 °C overnight, followed by the calcination in air at 300 °C for 3 h. Subsequently, to remove the residual chlorides, the sample was washed repeatedly with a dilute ammonia solution (1 mol L$^{-1}$), followed by filtrating and washing with deionized water until there was no more precipitate in the filtrate detected by AgNO$_3$ solution (0.1 mol L$^{-1}$). Finally, the sample was dried at 60 °C overnight. The sample thus obtained was denoted as the fresh Ru/TiO$_2$. Prior to catalytic performance tests, the fresh Ru/TiO$_2$ sample were reduced in situ under a H$_2$ gas flow (20 mL min$^{-1}$) at specific temperatures. The samples after reduction were denoted as Ru/TiO$_2$-$x$, where $x$ indicates the reduction temperature (200, 300, 400, 450, 500, or 600 °C). The Ru loading in the Ru/TiO$_2$ catalyst is 2.2 wt% as detected by ICP-OES.

**Catalyst characterization.** High-angle annular dark field scanning transmission electron microscopy (HAADF-STEM) together with the elemental mapping and high-resolution transmission electron microscopy (HRTEM) images were acquired using a JEOL JEM-2100F microscope operating at 200 kV. The Ru particle size was determined from HAADF-STEM images, and at least 200 particles were counted for each sample. The Ru concentrations in specimens were determined by inductively coupled plasma optical emission spectroscopy (ICP-OES) with an ICP-OES 7300DV instrument.

**CO and H$_2$ chemisorption experiments.** The exposure of Ru species after coating by TiO$_x$ was determined by CO and H$_2$ pulse chemisorption on a Micromeritics AutoChem II 2920 instrument. For CO (or H$_2$) chemisorption experiment, the sample (~100 mg) was pretreated with hydrogen at desired temperatures for 1 h, followed by purging with high-purity helium (or argon) for 30 min. After the sample was cooled down to 50 °C, a 5% CO/He (or 10% H$_2$/Ar) mixture was injected into the reactor repeatedly until CO (or H$_2$) adsorption was saturated. The dispersion of Ru was calculated from the amount of CO (or H$_2$) adsorbed by assuming the CO/Ru (or H/Ru) adsorption stoichiometry to be 1/1.

**X-ray absorption spectroscopy.** Pseudo-in situ X-ray absorption spectroscopy (XAS), including acquiring X-ray absorption near-edge structure (XANES) and extended X-ray absorption fine structure (EXAFS) data at the Ru K-edge, was performed at the BL 14W1 of Shanghai Synchrotron Radiation Facility (SSRF), China. A double Si (311) crystal monochromator was used for energy selection. The samples were pretreated in a H$_2$ flow (20 mL min$^{-1}$) at the desired tem-peratures for 2 h, followed by sealing with Capton film in a glove box without exposure to air. The spectra were collected at room temperature in the transition mode. The Ti $L_{3,2}$-edge XANES data was acquired at the XMCD beamline (BL12B)

at the National Synchrotron Radiation Laboratory (NSRL), China. The Athena software package was used to analyze the data.

**CO microcalorimetric measurements**. The differential heat of CO adsorption was measured for each specimen at 40 °C using a BT 2.15 Calvet calorimeter connected to gas handling and volumetric systems equipped with MKS 698A Baratron capacitance manometers ($\pm 1.33 \times 10^{-2}$ Pa). Previous to CO adsorption, the samples were treated in a $H_2$ flow at the desired temperatures for 1 h, followed by evacuation for 30 min at the same temperature. After cooling to room temperature in vacuum, the quartz tube was refilled with He and tightly sealed, after which the samples were outgassed at 40 °C overnight in the calorimetric cell. CO adsorption was carried out during the stepwise introduction of pure CO up to a pressure of ~10 Torr at 40 °C.

**In situ DRIFTS**. In situ diffuse reflectance infrared Fourier transform (DRIFT) spectra were acquired using a Bruker Equinox 55 spectrometer equipped with a mercury cadmium telluride (MCT) detector, recorded with a resolution of 4 cm$^{-1}$. Prior to CO adsorption, the samples were treated in situ in the DRIFT cell under a $H_2$ flow (20 mL min$^{-1}$) at the desired temperatures for 1 h, followed by purging with a He flow at the same temperature for 30 min. After cooling to 160 °C, a background spectrum was collected, following which the He flow was switched to a 5 vol% CO in He flow (20 mL min$^{-1}$) that was maintained until saturated adsorption was achieved. The system was purged with He to remove non-adsorbed CO and IR spectra were collected, such that CO adsorption data at 160 °C were obtained. After the treatment with He, the reactivity of adsorbed CO species ($CO_{ad}$) was monitored by switching to a 10 vol% $H_2$ in He flow (20 mL min$^{-1}$). Simultaneously, IR spectra were recorded every 30 s for 10 min. This experiment is referred to the reactivity of $CO_{ad}$ at 160 °C.

**Fischer–Tropsch tests**. FTS trials were performed in a stainless-steel fixed-bed reactor with an inner diameter of 12 mm under high pressure. Typically, the Ru/TiO$_2$ catalyst (20–40 mesh, 0.3 g) was diluted with quartz sand (20–40 mesh, 0.9 g) and loaded into the reactor. Prior to each reaction, the catalyst was reduced in a $H_2$ gas flow (20 mL min$^{-1}$) at the desired temperature (200–600 °C) for 2 h. After the reactor was cooled down, a syngas with a $H_2$/CO ratio of 2/1 ($H_2$/CO/Ar = 64/32/4 (v/v/v)) was introduced into the reactor at a flow rate of 15 mL min$^{-1}$ (space velocity = 3000 mL g$_{cat}^{-1}$ h$^{-1}$). Ar was used as an internal standard to calculate CO conversion and selectivity of CH$_4$ and CO$_2$. The reaction was carried out at 160 °C under 2.0 MPa. After passing through a hot trap (120 °C) and then an ice-bath, the gaseous products were analyzed online using an Agilent 7890 gas chromatograph equipped with an HP-PLOT/Q capillary column connected to a flame ionization detector (FID) and a TDX-01 column connected to a thermal conductivity detector (TCD). The data of the catalytic performances of Ru/TiO$_2$ catalysts were collected at the stable stage after at least 6 h of running. The calculation method for FTS catalytic performance was described in detail in the supplementary information.

**SSITKA experiments**. Steady-state isotopic transient kinetic analysis (SSITKA) is a combination of steady-state and transient techniques that can provide a reliable kinetic model to gain insights into the reaction mechanism. In this study, we used SSITKA to explore the activity of the Ru/TiO$_2$-$x$ catalysts with the aim of providing insights into the intrinsic promotional effects. During each SSITKA experiment, 50 mg of the sieved catalyst mixed with 200 mg of SiC was pretreated in situ in a $H_2$/Ar flow (20/20 mL min$^{-1}$) at the desired temperature (300, 450, or 600 °C). After cooling to 100 °C, the feed was switched to a $^{12}$CO/Ar mixture (35 mL min$^{-1}$, 0.85 bar CO, 1.77 bar inert). CO adsorption was determined by switching from $^{12}$CO/Ar to $^{13}$CO/Kr without changing the other reaction conditions. Subsequently, the feed was switched to $^{12}$CO/$H_2$/Ar (1.5/15/33.5 mL min$^{-1}$, 1.85 bar) for CO hydrogenation at 200 °C. After 6 h on stream, a switch from $^{12}$CO/$H_2$/Ar to $^{13}$CO/$H_2$/Kr was employed to study isotopic transients, while maintaining the CO conversion at ~10%. The isotopic transient response was determined by mass spectrometer.

The TOF values were calculated as

$$TOF_{CO} = \frac{R_{CO} \cdot M_{Ru}}{D \cdot x_{Ru}} \qquad (1)$$

and

$$TOF_{CH_4} = \frac{R_{CO} \cdot S_{CH_4} \cdot M_{Ru}}{D \cdot x_{Ru}} \qquad (2)$$

where $R_{CO}$ is the CO consumption in moles per gram of catalyst, $S_{CH4}$ is the methane selectivity, $M_{Ru}$ is the atomic mass of Ru, $D$ is the Ru dispersion and $x_{Ru}$ is the Ru loading of the sample.

The surface coverage of intermediates leading to CH$_4$ was calculated as

$$\theta_{CH_4} = \frac{N_{CH_4}}{N_{Total}} \qquad (3)$$

where $N_{CH4}$ is the number of intermediates leading to methane and $N_{Total}$ is the total number of active sites, as determined by the reversible adsorption of CO in the SSITKA experiments.

**Computational details**. Density functional theory (DFT) calculations were performed using the Vienna Ab-initio Simulation Package (VASP, a version of 5.4.4)[40,41]. The Perdew–Burke–Ernzerhof (PBE) exchange-correlation functional was used to initial calculation[42]. The core and valence electrons were represented by the projector augmented wave (PAW) potential, and the plane wave basis set with a cut-off energy of 500 eV was used. A TiO$_x$ cluster covered on a four-layered slab of the close-packed (001) surface derived from the hcp phase of Ru, was used as a model to the TiO$_x$ overlayer covered Ru nanocatalyst. A vacuum gap of 15 Å was used to separate periodic images of the slab in the direction perpendicular to the surface. The atoms in the top two layers of Ru(001) were allowed to relax during optimization. The Brillouin zone of the 4 × 4 surface unit cell of Ru(001) was sampled with a 2 × 2 × 1 Monkhorst-Pack grid. Optimized geometries were obtained by minimizing the forces on the atoms below 0.02 eV Å$^{-1}$. The transition state was first isolated using the climbing image nudged elastic band (CI-NEB) method and then refined using the dimer method to until force is below 0.02 eV Å$^{-1}$[43]. After that, the newly developed GW potential was adopted for the further optimization of adsorption geometries and transition states. The resulting transition state was finally confirmed by the normal mode frequency analysis, showing only one imaginary mode. We have first compared the relative stability of different TiO$_x$ ($x = 1-4$) clusters on the Ru(001) surface under different reduction degree condition which can be represented as the variation of chemical potential of oxygen, and was calculated according to the procedure of previous research[44]. In our calculation, the data of the formation energy of rutile TiO$_2$ was acquired from the reference[45]. Other computational details are shown in the Supplementary Information.

## Data availability

The data that support the findings of this study are available from the corresponding author upon reasonable request.

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

## Acknowledgements

This work was supported by the National Key R&D Program of China (2016YFA0202804), the Strategic Priority Research Program of the Chinese Academy of Sciences (XDB36030200), the National Natural Science Foundation of China (21978286, 21925803, 21776269), the Youth Innovation Promotion Association CAS.

## Author contributions

Y.Z., X.F.Y., X.S., and Y.H. conceived and designed the project. Y.Z. performed the experiments. X.F.Y. carried out the theoretical calculations. X.L.Y. and D.C. finished the SSITKA experiments. H.Q. performed the X-ray absorption experiments. Y.S. conducted the TEM observations. H.D., B.L.L., H.T., and B.L. contributed to the structure characterizations. Y.Z., X.F.Y., X.S., Y.H., and T.Z. analyzed the experimental data and prepared the paper. All authors reviewed and contributed to the paper.

## Competing interests

The authors declare no competing interests.
