## [Peer Review File · Nature Communications]

Reviewers' comments:

Reviewer #1 (Remarks to the Author):

The reviewer finds this paper interesting, but there are several areas in the paper where the reviewer thought that the presented work needed to be revised, which are noted below. However, even if these areas are addressed, the reviewer does not think that the work is sufficiently exciting to be published in Nature Communications, as discussed in point 1 below. In addition, the reviewer does not think that this paper will influence the thinking of the field since the modeling is not well described and the results that are presented are similar to what has been presented before [R1].

1. The presented work in Figure 3 does clearly demonstrate an enhancement of the catalytic activity. However, the reaction product (methane) is not very desirable. The results would be more compelling for such a prestigious journal if the reaction products would be more desirable, such as long-chain hydrocarbons or long-chain alcohols.
2. The authors interpret the decline in the peak intensity as indicative of an increasing degree of the reduction of TiO₂. However, such an analysis assumes that the oxidation state of the metal directly correlates with the peak intensity, which has been shown to not be the case [R2]. Indeed, the XANES spectra will also sensitively depend on the coordination environment as well. The XPS results also seem somewhat contradictory to the XANES results since an increased amount of Ti³⁺ is observed (which would seem to indicate that the Ti is getting oxidized and not reduced).
3. Previous work by Bell and coworkers [R1] have measured low wavenumber frequencies as evidence of C-O bond activation. However, looking at Figure 3 the referee sees no evidence of this in the measured spectra. As such, the referee does not see how the authors can claim that C-O bond activation occurs.
4. The authors argue that the bond cleavage of CO adsorbed on the Ru site at the interface becomes much more facile since the barrier is reduced from 2.07 eV to 1.60 eV. The referee would argue that such a barrier is still quite high.
5. The referee was very confused by the fact that the relative energy increased as the chemical potential increased in Figure 4a. This is possible, but only occurs in special cases (see [R3]). However, the authors do not give any equations in support of their claims. Such equations need to be given so that the reader may better understand what the origin of the result is, as shown in Figure 4a.
6. The configurations shown in Figure 4a that were used to construct the phase diagram are not given anywhere in the text nor in the SI. In particular, it is unclear what the configuration is for the O/Ru(001) structure. This information needs to be given.
7. There is additional information that needs to be given in the SI. In particular, no information is given

with regard to the underlying functional that was used in the calculation. Further, there are three versions of the PAW potentials that are currently available for the VASP code [R4]. The referee would kindly request that the version of the PAW potential be given in the text.

[R1] Johnson, G. R., Werner, S., & Bell, A. T. (2015). *ACS Catalysis*, 5(10), 5888–5903.

[R2] Kau, L., Spira-Solomon, D., Penner-Hahn, J., Hodgson, K., & Solomon, E. (1987). *Journal of the American Chemical Society*, 109(21), 6433–6442.

[R3] Reuter, K., & Scheffler, M. (2001). *Physical Review B*, 65(3), 035406.

[R4] Lejaeghere, K., Bihlmayer, G., Bjorkman, T., Blaha, P., Blügel, S., Blum, V., et al. (2016). *Science*, 351(6280), aad3000–aad3000

Reviewer #2 (Remarks to the Author):

Yaru Zhang and co-workers have studied in depth the regulation of SMSI in Ru/TiO₂ to enhance the performance in Fischer Tropsch synthesis (FT). The study excels in combining different techniques and approaches to deepen the understanding and to broaden the scope of the use of SMSI to steer the catalytic performance of metal nanoparticles on reducible support materials. These findings and the methodology applied is of importance to a large range of fields and scientists and is suitable for publication in a high-impact journal such as *Nature Communications*. Hereafter I provide suggestions to improve this paper even further.

1. In the Introduction the first paragraph addresses FT whereas both references 1 and 3 do not involve FT but rather OX-ZEO approaches to syngas conversion. The authors might consider to either change the references or the phrasing of their first sentence.
2. The paper involves control of SMSI to enhance catalyst performance. Most recently a review on precisely this topic has appeared in *Nature Catalysis* 2 (2019) 955–970. They might consider to add this reference to their paper.
3. From TEM images (e.g. Figure 1a,b) the authors conclude that Ru-NP are well dispersed on the support. However, the metal nanoparticles are very close together on the support in TEM images shown in the paper. From a broad brush calculation I find that this density of particles should lead to a Ru loading about one order of magnitude higher than that which is reported. I suggest that besides the TiO₂ particles covered by Ru-NP many other TiO₂ particles with hardly or no Ru NP are present. Low resolution TEM images or EDX might be required to (dis)prove this statement.
4. On page 5 the authors refer to ‘a visible coating of Ru-NP’. Please provide evidence.
5. For establishing the free Ru surface sites the authors have applied Cu-upd. The results in Table S3 reveal Ru dispersions between 4 and 8%. However, based on TEM particle size of about 2 nm a Ru dispersion of about 50% would be expected for the Ru/TiO₃-300 sample. For this sample one does not

expect SMSI to be significant. I suggest to use the more common technique of H₂ chemisorption to count Ru surface sites next to Cu-upd to unravel this inconsistency in the data.

6. For the XPS study I suggest to add the atomic ratio of Ru/Ti to Table S5. From that evidence for SMSI could be assessed since covering up Ru with TiO_x species should screen photo-electrons from Ru thus decreasing the Ru/Ti atomic ratio from XPS with increasing reduction temperatures.

7. The authors have used RuCl₃ as precursor for Ru-NP. My concern is that traces of Cl in the catalysts have affected the results and conclusions. Please provide XPS data on Cl in the samples. The Cl peak in XPS is expected to drop with increasing reduction temperature. Also the Cl/Ru atomic ratio will show whether or not this concern is relevant.

8. For the catalyst preparation (p. 16) please add the concentration of the RuCl₃ solution used. Also provide data on the precursor – purity etc. My experience with RuCl₃ is that sometimes large Ru particles are present in catalysts prepared therefrom.

9. Add data on Ru loading in the main text (e.g. on page 4); now one has to go deep into SI before one encounters the Ru loading of 2.2 wt% used throughout this study.

10. Have Ru loadings of the catalysts after thermal treatments been measured? Please add data. Please realize that calcination in air can lead to formation of RuO₄(g) and loss of Ru from the samples.

11. The calculation of TOF values (p. 21) should be redone after measuring H₂ chemisorption. The low Ru dispersions from Cu-upd give rise to very high TOF values but I doubt these to be accurate.

In summary, very good research that deserves publication after some modification.

Reviewer #3 (Remarks to the Author):

Manuscript title: Tuning Reactivity of Fischer-Tropsch Synthesis by Regulating TiO_x Overlayer for CO activation over Ru Nanocatalysts

Manuscript reference: NCOMMS-19-39843

This manuscript reports the catalytic performance of Ru/TiO₂ catalysts, pretreated at different reduction temperatures, in FT synthesis.

The manuscript includes the following sections: abstract, introduction, results (structural characterization, catalytic performance, catalytic mechanism, DFT calculation) and conclusion. A supplementary document is also available.

My opinion, questions and comments are as follows:

- The manuscript seemed to be too long for a communication.
- The title did not match well with the manuscript content: CO was not only activated over Ru nanocatalysts.
- SMSI seemed to be a key finding of this manuscript but H₂-TPR results showed that metal-support interaction was not « strong » in this catalytic system (Ru/TiO₂). Reduction temperatures were below 250°C and in the heterogeneous catalysis, this could not be considered as SMSI.
- Table S3: Unit of S_{sp} could not be so precise. How to explain the evolution of irregular dispersion versus reduction temperature ?

- The presence of TiO_x at the interface of Ru NPs and TiO₂ support surface is the crucial point (Fig. 1 f). Why HRTEM-EDX was not performed to highlight the presence of Ti on the surface of Ru NPs ?
- Table S6: comparing different catalytic systems under different conditions (in particular the reaction temperature) is not meaningful.
- Catalytic results: The catalytic deactivation was not reported. The evolution of the catalytic activity versus time was not communicated, which make the understanding of the results in Fig. 2 and Fig. S8 difficult. Were the results reported in these figures obtained at the beginning of the reaction or at the steady state ? If it is at the steady state, what was the evolution of the catalyst surface and was TiO_x phase still present on Ru particles ?
- In Fig. S8, at 200°C, the C₅₊ selectivity was ca. 60% which is mediocre in FT.
- The definition of CO conversion was wrong for a flow reactor. Inlet and outlet gas flow rates (of CO) must be employed, but not « quantity ». It is also the problem for different selectivity definitions.
- Any information on the analysis of the C₅₊ fraction was reported.
- Mass balance of around 90% showed a bad analytical practice.
- A major challenge of FT is its exothermicity. Support having a good thermal conductivity must be employed, but it is not the case of TiO₂-based materials. Why this choice of Ru/TiO₂ catalytic system instead of other supports which are more potential than TiO₂ ?

Doan Pham Minh

Reviewers' comments:

Reviewer #1:

The reviewer finds this paper interesting, but there are several areas in the paper where the reviewer thought that the presented work needed to be revised, which are noted below. However, even if these areas are addressed, the reviewer does not think that the work is sufficiently exciting to be published in Nature Communications, as discussed in point 1 below. In addition, the reviewer does not think that this paper will influence the thinking of the field since the modeling is not well described and the results that are presented are similar to what has been presented before [R1].

Question 1. *The presented work in Figure 3 does clearly demonstrate an enhancement of the catalytic activity. However, the reaction product (methane) is not very desirable. The results would be more compelling for such a prestigious journal if the reaction products would be more desirable, such as long-chain hydrocarbons or long-chain alcohols.*

Response:

The practical product distribution under the FT reaction conditions was presented in Figure R1. Notably, all Ru/TiO_{2-x} catalysts possess an excellent C₅₊ selectivity with a value up to 90% and only about 5% of CH₄ selectivity, indicating the promising application prospect of such Ru/TiO₂ in FTS for long-chain hydrocarbons, even though it possesses Ru NPs with small sizes (~2 nm) which generally result in a poor activity and C₅₊ selectivity in FTS [Carballo JMG, et al. *J Catal* **284**, 102–108 (2011); Kang J, Zhang S, Zhang Q, Wang Y. *Angew Chem Int Ed* **48**, 2565–2568 (2009)]^{1,2}.

In Figure 4a (Figure 3a before revising), in order to explain the unusual catalytic activity of Ru/TiO₂ in FTS, the kinetics toward CO activation was studied by the steady-state isotopic transient kinetic analysis (SSITKA), with which the evolution of intermediates with the associated coverage and reactivity can be acquired in Figure S16 (Figure S10 before revising). Here, in order to focus on the activity toward CO activation and to avoid the blockage of heavy hydrocarbons in the catalyst surface, the

feed gas of $H_2/CO = 10$ with a pressure of 0.185 MPa was employed in the SSITKA experiments. In this SSITKA characterization condition, the CH_4 selectivity will have an increase (Figure R1b), which then leads to a higher CH_4 selectivity than that under the FTS reaction conditions (Figure R1a). We have specified the SSITKA performing conditions in Figure 4a to make it clear.

Figure R1. (a) Catalytic performance of the Ru/TiO_{2-x} catalysts. Reaction conditions: 2 MPa, 160 °C, space velocity = 1200–6000 ml h⁻¹ g_{cat}⁻¹, H₂/CO/Ar = 64/32/4. (b) Catalytic Performance of the Ru/TiO_{2-x} catalysts in SSITKA experiments. Performing condition: 200 °C, 0.185 MPa, H₂/CO = 10.

Question 2. *The authors interpret the decline in the peak intensity as indicative of an increasing degree of the reduction of TiO₂. However, such an analysis assumes that the oxidation state of the metal directly correlates with the peak intensity, which has been shown to not be the case [R2]. Indeed, the XANES spectra will also sensitively depend on the coordination environment as well. The XPS results also seem somewhat contradictory to the XANES results since an increased amount of Ti³⁺ is observed (which would seem to indicate that the Ti is getting oxidized and not reduced).*

Response:

In Figure 2b (Figure 1e before revising) of our XANES results, the L_{3,2}-edge of Ti specimen, which corresponds to the p→d electron transition, has been measured in our experiment. In this case, the intensity of white-line peak was directly related to the unoccupied d-states of Ti specimen. As a result, the higher the white-line peak is, the

more the unoccupied d-orbitals Ti will possess. It thus provides us a tool to claim the oxidation states of Ti by comparing the peak intensity of metal *L*-edge of samples. From our result, the XANES spectra at the Ti *L*_{3,2}-edge exhibit a decline in peak intensities with the increase of pretreatment temperature, indicative of an increasing degree of the reduction of TiO₂.

While when using metal *K*-edge measurements of metal species [Hwang BJ, et al. *J Phys Chem B* **110**, 6475–6482 (2006); Shimizu K-i, Oda T, Sakamoto Y, Kamiya Y, Yoshida H, Satsuma A. *Appl Catal B* **111–112**, 509–514 (2012)]^{3, 4}, the white-line absorption is mainly attributed to the s→p electron transition. In this situation, the location of edge is generally used to determine the metal oxidation state as shown in Figure R2. This is because that the higher oxidation state of metal will cause a higher edge energy to excite the inner electrons. This is also the case in our XANES spectrum of Ru *K*-edge as shown in Figure S8 (Figure S6 before revising), it manifests as an energy shift towards lower energies by a decline of the oxidation state of Ru.

Figure R2. (a) Ru *K*-edge in situ XANES spectra for various reaction steps during the formation of Pt-Ru bimetallic NPs. The XANES patterns of reference compounds Ru powder and RuO₂ were also shown. Reprinted with permission from ref 3. Copyright 2006 American Chemical Society. (b) Rh *K*-edge XANES spectra for the representative samples. Reprinted with permission from ref 4. Copyright 2012

Elsevier.

As for the XPS results, the pristine sample of TiO₂ oxide support is mainly Ti⁴⁺ species. After reduction by H₂, the increased amount of Ti³⁺ was associated to the transformation of Ti⁴⁺ to Ti³⁺, which corresponds to the reduction of TiO₂. Hence, our XPS results are in good agreement with that of the XANES measurement of Ti L-edge.

Question 3. *Previous work by Bell and coworkers [R1] have measured low wavenumber frequencies as evidence of C-O bond activation. However, looking at Figure 3 the referee sees no evidence of this in the measured spectra. As such, the referee does not see how the authors can claim that C-O bond activation occurs.*

Response:

As shown in the *in situ* DRIFT spectra of Figure 4e (Figure 3e before revising), the conversion of CO to methane occurs with a consumption of Ruⁿ⁺(CO)_x (2136 cm⁻¹) and Ruⁿ⁺-CO (2075 cm⁻¹). However, the further conversion of CO was restrained in the absence of Ruⁿ⁺(CO)_x and Ruⁿ⁺-CO. It then leaves the Ru_x-CO (2035 cm⁻¹) as a predominant CO chemisorption on the Ru surface. Therefore, the interface of partially oxidized Ruⁿ⁺ sites were supposed to be the active sites for the FTS reaction in the Ru/TiO₂-450 catalysts. Such evolution of CO species was also observed on the other Ru/TiO₂-*x* catalysts, including the Ru/TiO₂-300 and the Ru/TiO₂-600 samples (Figure S18).

Meanwhile, as shown in Figure R3, the intensity of CO related to FTS on the Ru/TiO₂-450 was more remarkable than that of Ru/TiO₂-300 and the Ru/TiO₂-600 catalysts, which was responsible for its higher activity in FTS. In this regard, the activation pattern of CO on the Ru/TiO₂-*x* catalysts might differ from the previous report on the Co-Mn catalysts by Bell and coworkers [*Johnson GR, Werner S, Bell AT. ACS Catal 5, 5888–5903 (2015)*]⁵, in which a low wavenumber frequency was measured as evidence of C-O bond activation. Furthermore, with DFT calculations in Figure 5b, the TiO_x overlayer on the Ru was found to be directly involved into the

cleavage of C–O bond by lowering the energy barrier of 0.47 eV, indicating that the interface with the TiO_x will promote the CO activation on our $\text{Ru}/\text{TiO}_{2-x}$ catalyst.

Figure R3. *In situ* DRIFT spectra obtained after CO adsorption and evacuation with helium at 160 °C, over the $\text{Ru}/\text{TiO}_{2-x}$ catalysts.

The corresponding discussion has been added in the revised manuscript.

Question 4. *The authors argue that the bond cleavage of CO adsorbed on the Ru site at the interface becomes much more facile since the barrier is reduced from 2.07 eV to 1.60 eV. The referee would argue that such a barrier is still quite high.*

Response:

In our model calculation, we have first estimated the CO bond cleavage on the Ru(001) surface. As seen in Figure 5b (Figure 4b before revising), it has an energy barrier as high as 2.15 eV, which was in a good agreement with the previous report of 227 kJ/mol [Ciobica IM, van Santen RA. *J Phys Chem B* **107**, 3808–3812 (2003)]⁶. The Ru(001) surface decorated by the TiO_x cluster was used to model the interface of TiO_x overlayer on Ru NPs (Figure 5a). By decreasing the oxygen chemical potential under reduction condition, the reduction of TiO_4 occurred readily on the Ru surface

through a sequential reduction to $\text{TiO}_3/\text{Ru}(001)$ and $\text{TiO}_2/\text{Ru}(001)$, respectively. Meanwhile, the reduction of $\text{TiO}_4/\text{Ru}(001)$ to $\text{TiO}_3/\text{Ru}(001)$ was even thermodynamically more favorable than the surface reduction of O adspecies on the parent $\text{Ru}(001)$ surface, indicative of the facile formation of reduced TiO_x on the $\text{Ru}(001)$ surface. This was consistent with the experimental observation of a reduction of the TiO_x overlayer under the reduction condition.

With the model of TiO_3 cluster decorating on the $\text{Ru}(001)$ surface, the bond cleavage of CO adsorbed on the Ru site of interface becomes much facile by experiencing a calculated barrier of 1.62eV (Figure 5b), with the aid of TiO_3 as the O seizer of carbonyl group to transform to TiO_4 . As a result, the TiO_x overlayer on the Ru promotes the C–O bond cleavage by lowering the energy barrier of 0.47 eV, indicating that the interface with the direct involvement of TiO_x will facilitate the CO activation on our $\text{Ru}/\text{TiO}_{2-x}$ catalyst, which was also suggested by Bell and coworkers [*Johnson GR, Werner S, Bell AT. ACS Catal 5, 5888–5903 (2015)*]⁵. Meanwhile, taking into account that our experiments of FTS were conducted at a reaction temperature of 160–200 °C and a reaction pressure of 2 MPa, such a barrier is facile to overcome on the Ru/TiO_2 catalysts at the reaction condition of FTS.

In the revised manuscript, the corresponding discussion has been added.

Question 5. *The referee was very confused by the fact that the relative energy increased as the chemical potential increased in Figure 4a. This is possible, but only occurs in special cases (see [R3]). However, the authors do not give any equations in support of their claims. Such equations need to be given so that the reader may better understand what the origin of the result is, as shown in Figure 4a.*

Response:

Thank you. We have provided the calculation details in the supplementary information of the revised version.

The relative stability of different TiO_x ($x = 1-4$) clusters on the $\text{Ru}(001)$ surface under different reduction degree conditions which can be represented as the variation of chemical potential of oxygen, was calculated according to the procedure of

previous research [Reuter K, Scheffler M. *Phys Rev B* **65**, 035406 (2001)]⁷.

Considering a successive reduction of TiO₄/Ru(001) to TiO₂/Ru(001), the energy of removing an oxygen (ΔE_r) on TiO_x/Ru(001) can be expressed as

$$\Delta E_r = E(\text{TiO}_{x-1}) + \mu_o - E(\text{TiO}_x)$$

Here, the chemical potential of O atom (μ_o) is restrained between

$$\mu_{o_2} + 1/2 H_f(\text{TiO}_2) < \mu_o < \mu_{o_2}$$

$$\mu_{o_2} = 1/2 E(\text{O}_2)$$

due to the limitation of the non-condensed condition of Ti metal and O₂ solid on our Ru/TiO_x catalysts, which corresponds to the O-poor and O-rich conditions, respectively. μ_{o_2} refers to the chemical potential of gaseous O₂. $E(\text{O}_2)$ is the total energy of a free O₂ molecule, and the data of the formation energy of rutile TiO₂ ($H_f(\text{TiO}_2)$) was acquired from the reference (-10.30 eV) [Lide DR. *CRC Handbook of Chemistry and Physics, 88th Edition. Taylor & Francis Group, Boca Raton, (2007)*]⁸.

The most preferable bonding geometry of TiO_x clusters (TiO₄, TiO₃, and TiO₂) on the Ru(001) was determined in thermodynamics, with their corresponding geometries shown in Figure R4 and the relative energy of successive reduction steps under O-rich and O-poor condition in Table R1.

Figure R4. Thermodynamic stability of different TiO_x/Ru(001) and O/Ru(001) under a variation of the chemical potential of O, with referring to TiO₃/Ru(001) and Ru(001), respectively, with the atomic configuration in insets.

Table R1. Relative energy of successive reduction steps under O-rich and O-poor condition.

Reduction step	O-rich (eV)	O-poor (eV)
$\text{TiO}_4 \rightarrow \text{TiO}_3 + \text{O}$	2.66	-2.49
$\text{TiO}_3 \rightarrow \text{TiO}_2 + \text{O}$	2.94	-2.21
$\text{O/Ru(001)} \rightarrow \text{Ru(001)} + \text{O}$	2.87	-2.28

In Figure R4, we have taken the $\text{TiO}_3/\text{Ru(001)}$ as a reference for its further oxidation to $\text{TiO}_4/\text{Ru(001)}$ or reduction to $\text{TiO}_2/\text{Ru(001)}$. The further oxidation of $\text{TiO}_3/\text{Ru(001)}$ to $\text{TiO}_4/\text{Ru(001)}$ will lead to its relative energy with $\text{TiO}_4/\text{Ru(001)}$ getting more negative as the oxygen chemical potential increasing. While for the reduction of $\text{TiO}_3/\text{Ru(001)}$ with O atom removed to $\text{TiO}_2/\text{Ru(001)}$, the relative energy will increase with increasing the chemical potential of O_2 .

Similar procedure was also proceeded to estimate the reduction of O/Ru(001) to Ru(001) under the same chemical potential condition. For the O/Ru(001) surface, oxygen atom in a 4×4 supercell with a O-coverage of $1/16$ was considered in the calculation of binding energy of O adatom on Ru(001) surface, by taking the Ru(001) surface as a reference in Figure R4.

Question 6. *The configurations shown in Figure 4a that were used to construct the phase diagram are not given anywhere in the text nor in the SI. In particular, it is unclear what the configuration is for the O/Ru(001) structure. This information needs to be given.*

Response:

Thank you. The most preferable binding geometry of TiO_x clusters (TiO_4 , TiO_3 , and TiO_2) on the Ru(001) was determined in thermodynamics, and the corresponding configurations have been added as insets in Figure 5a (Figure 4a before revising). For the O/Ru(001) surface, oxygen atom in a 4×4 supercell with a coverage of $1/16$ was taken as a model for us to estimate the binding energy of O adatom on the Ru(001)

surface, by taking the Ru(001) surface as a reference in Figure 5a.

Question 7. *There is additional information that needs to be given in the SI. In particular, no information is given with regard to the underlying functional that was used in the calculation. Further, there are three versions of the PAW potentials that are currently available for the VASP code [R4]. The referee would kindly request that the version of the PAW potential be given in the text.*

Response:

Thank you for your suggestion. The more detailed information with regard to the underlying functional has been added in the DFT calculation section in the revised supplementary information.

Our relativistic DFT calculations were performed using the VASP code (a version of 5.4.4). The Perdew-Burke-Ernzerhof (PBE) exchange-correlation functional was used. The core and valence electrons were represented by the projector augmented wave (PAW) potential, and the plane wave basis set with a cut-off energy of 500 eV was used. The core and valence electrons were represented by the projector augmented wave potential updated in 2012 (potpaw_PBE.5.2), which has been proved to acquire a reliable chemical accuracy in solid calculations [*Lejaeghere K, et al. Science 351, aad3000 (2016)*]⁹. The valence electrons were designated of Ti ($3d^34s^1$), O ($2s^22p^4$), Ru ($4d^75s^1$), and C ($2s^22p^2$) for the initial geometry searching and transition state locating. Optimized geometries were obtained by minimizing the forces on the atoms below $0.02 \text{ eV } \text{\AA}^{-1}$. The transition state was first isolated using the climbing image nudged elastic band (CI-NEB) method and then refined using the dimer method to until force is below $0.02 \text{ eV } \text{\AA}^{-1}$. The resulting transition state was finally confirmed by the normal mode frequency analysis, showing only one imaginary mode. After that, the newly developed GW potential in potpaw_PBE.5.2, with the valence electronic configuration of Ti ($3s^23p^63d^4$), O ($2s^22p^4$), Ru ($4s^24p^64d^8$), and C ($2s^22p^2$) was adopted for the further optimization of adsorption geometries and transition states.

From DFT results in Figure R5, these two different types of PAW potential result in

comparable results. In Figure 5, we have updated our results obtained from GW potential calculation in the revised manuscript.

Figure R5. Thermodynamic stability of different TiO_x/Ru(001) and O/Ru(001) under a variation of the chemical potential of O, and possible catalytic mechanisms of CO activation at PBE (a, b) and GW (c, d) level calculations on the TiO₃/Ru(001) model surface (red line), with the dissociation of CO on Ru(001) surface as a comparison (blue line).

Reviewer #2:

Yaru Zhang and co-workers have studied in depth the regulation of SMSI in Ru/TiO₂ to enhance the performance in Fischer Tropsch synthesis (FT). The study excels in combining different techniques and approaches to deepen the understanding and to broaden the scope of the use of SMSI to steer the catalytic performance of metal nanoparticles on reducible support materials. These findings and the methodology applied is of importance to a large range of fields and scientists and is suitable for publication in a high-impact journal such as Nature Communications. Hereafter I provide suggestions to improve this paper even further.

Question 1. *In the Introduction the first paragraph addresses FT whereas both references 1 and 3 do not involve FT but rather OX-ZEO approaches to syngas conversion. The authors might consider to either change the references or the phrasing of their first sentence.*

Response:

Thank you. The references 1 and 3 has been replaced by the two recently published references on the FTS researches (Zhong L, *et al.* Cobalt carbide nanoprisms for direct production of lower olefins from syngas. *Nature* **538**, 84-87 (2016); Torres Galvis HM, Bitter JH, Khare CB, Ruitenbeek M, Dugulan AI, de Jong KP. Supported iron nanoparticles as catalysts for sustainable production of lower olefins. *Science* **335**, 835–838 (2012)).

Question 2. *The paper involves control of SMSI to enhance catalyst performance. Most recently a review on precisely this topic has appeared in Nature Catalysis 2 (2019) 955–970. They might consider to add this reference to their paper.*

Response:

The most recent review (van Deelen TW, Hernández Mejía C, de Jong KP. Control of metal-support interactions in heterogeneous catalysts to enhance activity and selectivity. *Nat Catal* **2**, 955-970 (2019)) has been cited in our revised manuscript (Reference 21).

Question 3. *From TEM images (e.g. Figure 1a, b) the authors conclude that Ru-NP are well dispersed on the support. However, the metal nanoparticles are very close together on the support in TEM images shown in the paper. From a broad brush calculation, I find that this density of particles should lead to a Ru loading about one order of magnitude higher than that which is reported. I suggest that besides the TiO₂ particles covered by Ru-NP many other TiO₂ particles with hardly or no Ru NP are present. Low resolution TEM images or EDX might be required to (dis)prove this statement.*

Response:

Thank you for your suggestion. We have estimated the density of metal nanoparticles by counting the NPs on the STEM images in Figure 1b (Figure 1a before revising). It was estimated to be ~ 0.03 NPs/nm².

On the other hand, as for a surface area of $36 \text{ m}^2 \text{ g}^{-1}$ and a metal loading of 2.2 wt% Ru/TiO₂, the density of NPs can be obtained according to the equations by considering the hemispheric shape of a 1.7 nm Ru on the support:

$$N \approx 1/2 \times (D/d)^3 = 0.5 \times (1.7/0.265)^3 \approx 132 \text{ atoms/NP},$$

where N represents the number of metal atom per nanoparticle, with D as the size of metal NPs, and d refers to the diameter of a single Ru atom.

Then the theoretical NP density (n) on the support can be calculated by

$$n \approx (m/M) \times N_A/S/N = (2.2\%/101.07) \times 6.02 \times 10^{23}/(36 \times 10^{18})/132 \approx 0.028 \text{ NPs/nm}^2,$$

where m is the loading of metal, and M is the atomic weight of metal Ru, with the Avogadro constant (N_A) and the surface area of the sample (S) in above equation.

From our results, the observation of the size distribution is close to the theoretical density of particle, which suggests a homogenous distribution of Ru NPs on the TiO₂ support.

Experimentally, as suggested by the referee, low resolution STEM images of the Ru/TiO₂ catalyst and the elemental mapping have been explored to confirm the high dispersion of Ru NPs on the support. As shown in Figure R6, the Ru nanoparticles are

evenly distributed on the TiO_2 support. This can be attributed to the intimate interaction between RuO_2 and rutile TiO_2 , which in turn stabilizes the Ru NPs and maintains its high dispersion.

Figure R6. (a, b) Low resolution HAADF-STEM images of the fresh Ru/TiO_2 catalyst. (c) Elemental mapping of Ru/Ti/O in the fresh Ru/TiO_2 catalyst.

Question 4. *On page 5 the authors refer to ‘a visible coating on Ru-NP’. Please provide evidence.*

Response:

We have performed HRTEM characterizations on Ru NPs located at the edge of support. From the images of the Ru/TiO_2 -500 and Ru/TiO_2 -600 samples in Figure R7, an atomic packing of Ru atoms can be distinguished on Ru NPs. Furthermore, a thin but visible coating with an indistinct boundary was readily observed over Ru NPs, indicating the capsulation of NPs by TiO_2 due to the SMSI of Ru/TiO_2 . The corresponding discussion has been added in the revised manuscript.

Figure R7. HRTEM images of the Ru/TiO₂-500 and Ru/TiO₂-600 samples.

Question 5. *For establishing the free Ru surface sites, the authors have applied Cu-upd. The results in Table S3 reveal Ru dispersions between 4 and 8%. However, based on TEM particle size of about 2 nm a Ru dispersion of about 50% would be expected for the Ru/TiO₂-300 sample. For this sample one does not expect SMSI to be significant. I suggest to use the more common technique of H₂ chemisorption to count Ru surface sites next to Cu-upd to unravel this inconsistency in the data.*

Response:

Thank you for your suggestion. Owing to the presence of TiO_x coating on Ru NPs after reduction pretreatment, the determination of Ru dispersion using TEM particle size will lead to an overestimation of metal dispersions, especially for the samples pretreated at high temperature with a great amount of TiO_x coating. According to the suggestion of the reviewer, the dispersions of Ru for different Ru/TiO₂-x samples were also determined by H₂ and CO pulse chemisorption, where the H/Ru or CO/Ru adsorption stoichiometry was assumed to be 1/1. The corresponding results were displayed in Table R2.

As we can see, the values obtained by Cu upd, H₂ chemisorption and CO chemisorption give the same tendency of the metal dispersions for different Ru/TiO₂-x samples, that is, the dispersion of Ru decreases with increasing the reduction temperature from 300 to 600 °C. As compared, Cu upd and H₂ chemisorption show a lower Ru dispersion than that of CO chemisorption. It might be caused by the Ruⁿ⁺

sites at the Ru-TiO₂ interface, which are unavailable for Cu deposition and H₂ chemisorption, but it can be readily involved in the CO chemisorption, as indicated by our observation with *in situ* DRIFT spectra (Figure R3).

Table R2. Cu upd, H₂ and CO pulse chemisorption results for the Ru/TiO_{2-x} catalysts.

Temp. (°C)	Area by Cu upd (cm ²)	D _{Ru} by upd (%)	H ₂ uptake (μmol g ⁻¹)	D _{Ru} by H ₂ (%)	CO uptake (μmol g ⁻¹)	D _{Ru} by CO (%)
200	0.300	–	28.1	25.8	102.6	47.2
300	0.405	8.1	32.2	29.6	94.2	43.3
400	–	–	27.5	25.3	84.7	38.9
450	0.202	4.0	21.6	19.9	74.0	34.0
500	–	–	17.4	16.0	59.8	27.5
600	0.189	3.8	10.8	9.9	38.0	17.5

Figure R8. Reaction rates and TOF values for the Ru/TiO_{2-x} catalysts.

Correspondingly, we have calculated the TOF in FTS by using the Ru dispersion determined from CO chemisorption, and the results were listed in Table S6. The variation in TOF value also exhibits a volcano-type trend with increasing the

pretreatment temperature from 300 to 600 °C (Figure R8).

In our revised manuscript, the TOF has been updated by using the dispersion data for CO chemisorption, and the corresponding discussion has also been added.

Question 6. *For the XPS study I suggest to add the atomic ratio of Ru/Ti to Table S5. From that evidence for SMSI could be assessed since covering up Ru with TiO_x species should screen photo-electrons from Ru thus decreasing the Ru/Ti atomic ratio from XPS with increasing reduction temperatures.*

Response:

Thank you for your suggestion. The atomic ratio of Ru/Ti has been added in Table S5. The Ru/Ti atomic ratios show a successive decline from 12.9% to 11.6%, then to 10.7% with increasing reduction temperatures from 300 to 450, then to 600 °C, which in turn demonstrated the gradual encapsulation of TiO_x species on Ru NPs.

Question 7. *The authors have used RuCl₃ as precursor for Ru-NP. My concern is that traces of Cl in the catalysts have affected the results and conclusions. Please provide XPS data on Cl in the samples. The Cl peak in XPS is expected to drop with increasing reduction temperature. Also, the Cl/Ru atomic ratio will show whether or not this concern is relevant.*

Response:

As shown in Figure R9, no Cl (with an absorption peak at around 198 eV [McEvoy *AJ. phys stat sol (a)* **71**, 569-574 (1982); Pollini II. *Phys Rev B* **50**, 2095-2103 (1994); Morgan DJ. *Surf Interface Anal* **47**, 1072-1079 (2015)]^{10, 11, 12}) was detected in XPS spectra for the Ru/TiO_{2-x} catalysts. This was because that, the residual chlorides after calcination have been removed during the procedure of repeatedly washing with ammonia solution (1 mol L⁻¹), until there was no Cl in the filtrate (detected by AgNO₃ solution of 0.1 mol L⁻¹).

Figure R9. XPS data of the Ru/TiO₂-x samples.

Question 8. For the catalyst preparation (p. 16) please add the concentration of the RuCl₃ solution used. Also provide data on the precursor – purity etc. My experience with RuCl₃ is that sometimes large Ru particles are present in catalysts prepared therefrom.

Response:

The concentration of the RuCl₃·3H₂O (AR) solution is 8.2 wt% of metal Ru, with 0.0317 g Ru per gram of solution. As shown in Figure R10a, the RuCl₃ precursor can be highly dispersed on TiO₂ upon impregnation. Due to the high degree of lattice matching between rutile TiO₂ and RuO₂, the RuO₂ tend to spread and stabilize on the TiO₂ support instead of aggregation upon calcination in air, and the Ru NPs remain highly dispersed even after high temperature reduction (Figure R10b). This point can be further confirmed that no large Ru particles were observed in the low resolution STEM images (Figure R6).

Figure R10. HAADF-STEM images of Ru/TiO₂ samples upon different steps. (a) Fresh RuCl₃/TiO₂ catalyst after impregnation and drying overnight. (b) The obtained Ru/TiO₂-500 catalyst after thermal treatment in air at 300 °C followed by reduction in H₂ at 500 °C.

Question 9. *Add data on Ru loading in the main text (e.g. on page 4); now one has to go deep into SI before one encounters the Ru loading of 2.2 wt% used throughout this study.*

Response:

Done. We have added data on Ru loading (2.2 wt%) in the main text for the Ru/TiO₂ catalyst.

Question 10. *Have Ru loadings of the catalysts after thermal treatments been measured? Please add data. Please realize that calcination in air can lead to formation of RuO₄(g) and loss of Ru from the samples.*

Response:

Done. The loading of Ru, after calcination in air and the following chlorides removal process, has been measured and determined to be 2.2 wt% by ICP-OES. Due to the lattice matching between rutile TiO₂ and RuO₂, the Ru precursor (RuCl₃) dominantly transformed into RuO₂ in the calcination process, and the strong interaction between RuO₂ and TiO₂ avoids the formation of RuO₄(g) and possible loss of Ru during the calcination.

Question 11. *The calculation of TOF values (p. 21) should be redone after measuring H₂ chemisorption. The low Ru dispersions from Cu-upd give rise to very high TOF values but I doubt these to be accurate.*

Response:

As stated in the response to Question 5, we have calculated the TOF in FTS by using the Ru dispersion determined from CO chemisorption, and the results were listed in Table S6. The variation in TOF value also exhibits a volcano-type trend with increasing the pretreatment temperature from 200 to 600 °C (Figure R8).

In summary, very good research that deserves publication after some modification.

Reviewer #3:

Manuscript title: Tuning Reactivity of Fischer-Tropsch Synthesis by Regulating TiO_x Overlayer for CO activation over Ru Nanocatalysts

Manuscript reference: NCOMMS-19-39843

This manuscript reports the catalytic performance of Ru/TiO₂ catalysts, pretreated at different reduction temperatures, in FT synthesis.

The manuscript includes the following sections: abstract, introduction, results (structural characterization, catalytic performance, catalytic mechanism, DFT calculation) and conclusion. A supplementary document is also available.

My opinion, questions and comments are as follows:

Question 1. *The manuscript seemed to be too long for a communication.*

Response:

The length of article can range from short communications to more in-depth studies in *Nature Communication*, and the length of this article meet the requirements of *Nature Communications* well.

Question 2. *The title did not match well with the manuscript content: CO was not only activated over Ru nanocatalysts.*

Response:

According to your suggestion, we have modified the title to “Tuning reactivity of Fischer–Tropsch synthesis by regulating TiO_x overlayer over Ru/TiO₂ nanocatalysts”.

Question 3. *SMSI seemed to be a key finding of this manuscript but H₂-TPR results showed that metal-support interaction was not « strong » in this catalytic system (Ru/TiO₂). Reduction temperatures were below 250 °C and in the heterogeneous catalysis, this could not be considered as SMSI.*

Response:

The H₂-TPR profile (Figure R11) of the Ru/TiO₂ catalyst displays three main peaks, ascribed to the reduction of RuO₂ species with different interfacial interactions with

the TiO₂. The peaks (at 189 and 208 °C) show higher temperatures than those in Ru/Al₂O₃ (at 157 °C) and Ru/SiO₂ (at 146 °C) catalysts, demonstrating the presence of chemical interactions between the RuO₂ and rutile TiO₂ due to the lattice match of oxides. Consequently, the Ru/TiO₂ catalyst was stable so as to avoid particle growth during reduction.

Figure R11. H₂-TPR profile obtained from the Ru-based catalyst on TiO₂, Al₂O₃ and SiO₂ supports.

More importantly, a less intense, broad peak also appears between 300 and 800 °C, which was attributed to reduction of the TiO₂ support due to H₂ spillover from the Ru to the TiO₂, with TiO_x overlayer covering the Ru NPs. According to the definition of strong metal–support interaction (SMSI) by Tauster [Tauster SJ, Fung SC, Garten RL. *J Am Chem Soc* **100**, 170-175 (1978); Tauster SJ, Fung SC, Baker RTK, Horsley JA. *Science* **211**, 1121-1125 (1981)]^{13, 14}, which demonstrates the presence of SMSI between Ru and rutile TiO₂. In fact, various noble metals in group VIII (8–10) were suggested to be involved into an SMSI when these substances were supported on titanium oxide [Tauster SJ. *Acc Chem Res* **20**, 389-394 (1987)]¹⁵.

In this case, the reduction of Ru/TiO₂ at a temperature higher than 300 °C will cause a SMSI behavior of Ru on TiO₂, that is, TiO_x overlayer begins to encapsulate

Ru NPs and acquires a varied catalytic performance in FTS.

Question 4. *Table S3: Unit of S_{sp} could not be so precise. How to explain the evolution of irregular dispersion versus reduction temperature?*

Response:

Cu upd technique is usually used to detect the surface area (S_{sp} , m^2/g) of bulk metal or metal alloys¹⁶. The only 2.2 wt% Ru loading in the Ru/TiO₂ catalysts poses a great challenge to the determination of Ru surface area by Cu upd, which causes the quantitative data not precise enough. Moreover, Cu upd shows a much lower dispersion than that of CO chemisorption (Table R2). It might be caused by the Ruⁿ⁺ sites at the Ru-TiO₂ interface, which are unavailable for the method of Cu upd. Nevertheless, it can still be utilized as an effective tool to get a qualitative comparison of the exposure of metallic Ru for different Ru/TiO_{2-x} catalysts.

In Cu upd process, we assume that a single Cu atom deposits on one surface Ru to form a monolayer deposition, which can be realized by judicious choice of electrochemical potential and deposition time.

The specific surface area of metallic Ru (i.e. S_{sp} of Ru, m^2/g) can be calculated by the integration of the peak area corresponding to upd stripping, which can further be used to acquire the exposure of surface metallic Ru.

It is worth noting that only the metallic Ru species in reduction state can act as deposition site for Cu upd. In contrast, the Ru species in oxidation state are inert site for Cu upd. Therefore, only the amount of surface metallic Ru can be acquired from Cu upd experiments, which in turn can provide the residual metallic Ru sites after covering by TiO_x overlayers. From Cu upd results (Figure S7), the exposure of surface metallic Ru shows a decline with increasing the reduction temperature from 300 to 600 °C, which was caused by the gradual encapsulation of TiO_x coating on Ru NPs. While the unusual lower Ru⁰ exposure of Ru/TiO₂-200 than the Ru/TiO₂-300 sample was caused by the incomplete reduction of surface RuO₂ under 200 °C, which is in good agreement with our H₂-TPR experiments (Figure S6).

Question 5. *The presence of TiO_x at the interface of Ru NPs and TiO₂ support surface is the crucial point (Fig. 1 f). Why HRTEM-EDX was not performed to highlight the presence of Ti on the surface of Ru NPs?*

Response:

Thank you for your suggestion. We have performed HRTEM characterizations on Ru NPs located at the edge of support. From the images of the Ru/TiO₂-500 and Ru/TiO₂-600 samples in Figure R7, an atomic packing of Ru atoms can be distinguished on Ru NPs. Furthermore, a thin but visible coating with an indistinct boundary was readily observed over Ru NPs, indicating the capsulation of NPs by TiO₂ due to the SMSI of Ru/TiO₂. The corresponding discussion has been added in the revised manuscript. It is unavailable to study the TiO_x coating by using the EDX technique, as the Ru/TiO_{2-x} samples exhibit a size distribution with a diameter less than 2 nm.

Question 6. *Table S6: comparing different catalytic systems under different conditions (in particular the reaction temperature) is not meaningful.*

Response:

The raising of reaction temperature will result in a significant enhancement of activity in FTS. As shown in Table S7 (Table S6 before revising), even though the Ru/TiO₂ catalyst in this work was performed at a lower temperature (160 °C) than the reports of references (> 200 °C), it can still achieve a high activity in FTS. It thus suggests the decent performance of such Ru/TiO₂ catalyst in FTS under mild conditions.

Question 7. *Catalytic results: The catalytic deactivation was not reported. The evolution of the catalytic activity versus time was not communicated, which make the understanding of the results in Fig. 2 and Fig. S8 difficult. Were the results reported in these figures obtained at the beginning of the reaction or at the steady state? If it is at the steady state, what was the evolution of the catalyst surface and was TiO_x phase still present on Ru particles?*

Response:

A typical evolution of catalytic performance versus time over the Ru/TiO_{2-x} catalysts were shown in Figure R12a, b. The data of the catalytic performances of Ru/TiO_{2-x} catalysts were collected at the stable stage after at least 6 hours' running. HRTEM image of the Ru/TiO₂-600-spent catalyst was also explored and shown in Figure R12d, the TiO_x overlayer remained on the Ru nanoparticles after 24 hours testing, indicative of the importance of TiO_x overlayer in determining the catalytic activity of Ru in FTS. Furthermore, the recycled Ru/TiO₂-450-spent and Ru/TiO₂-600-spent catalysts without any regeneration treatment were still active for FTS (Figure R12e, f), which in turn offers the great possibility of the presence of TiO_x overlayer on the catalyst surface during the reaction process.

Figure R12. Evolution of catalytic performance versus time over the (a) Ru/TiO₂-450, (b) Ru/TiO₂-600 catalyst. (c, d) HRTEM images of the fresh Ru/TiO₂-600 and the

Ru/TiO₂-600-spent catalyst (Ru/TiO₂-600 after 24 hours testing). (e, f) Catalytic FT performance over the recycled Ru/TiO₂-450-spent and Ru/TiO₂-600-spent catalyst without any regeneration treatment.

Question 8. *In Fig. S8, at 200 °C, the C₅₊ selectivity was ca. 60% which is mediocre in FT.*

Response:

We agree with reviewer that the C₅₊ selectivity was ca. 60% which is mediocre in FT at 200 °C as shown in Figure S10 (Figure S8 before revising). As the reaction temperature increased, the CO conversion together with selectivity of light hydrocarbons (CH₄ and C₂–C₄) would have a rapid increase in FTS. In order to maintain similar CO conversion (an increased CO conversion would pose a great challenge to heat transfer as FTS reaction is highly exothermic), we increased the GHSV from 3000 mL h⁻¹ g_{cat}⁻¹ to 9000 mL h⁻¹ g_{cat}⁻¹ when FTS reaction was conducted at 200 °C, which would further decrease the C₅₊ selectivity. Hence, the increasing temperature from 160 to 200 °C together with the increasing GHSV leads to a decreased selectivity of C₅₊ from ~90% to ~60 % in FTS.

Question 9. *The definition of CO conversion was wrong for a flow reactor. Inlet and outlet gas flow rates (of CO) must be employed, but not « quantity ». It is also the problem for different selectivity definitions.*

Response:

We agree with reviewer that CO conversion together with the product selectivity should employ the gas flow rates in the inlet and outlet. We provided a conceptual definition for CO conversion and the product selectivity in the first version of manuscript. In fact, we used data detected by gas chromatograph (GC), which equipped with a TCD (concentration sensitive detector) and an FID (mass flow rate sensitive detector), to determine the CO conversion and product selectivity. For the variable volume in Fischer–Tropsch synthesis reaction, the inert Ar was used as an internal standard to calculate CO conversion and product selectivity of CH₄ and CO₂.

We have added the information about the calculation method in the revised supplementary information, which is described in detail as follows.

The feed gas (H₂/CO/Ar) and the gaseous products (including CO₂, CH₄ and C₂–C₄ hydrocarbons) were analyzed online by gas chromatograph (GC). The catalytic results were determined by the peak areas of the components identified by GC which was equipped with an HP-PLOT/Q capillary column connected to a flame ionization detector (FID) and a TDX-01 column connected to a thermal conductivity detector (TCD).

The thermal conductivity detector (TCD) was used to detect inorganic gaseous, including Ar, CO, CH₄ and CO₂. The CO conversion, CH₄ selectivity and CO₂ selectivity can be determined by the peak areas of the components identified by TCD.

The CO conversion, X_{CO} , was calculated using the equation

$$X_{CO} = \frac{n_{in}(CO) - n_{out}(CO)}{n_{in}(CO)} = 1 - \frac{A_{out}(CO)/A_{out}(Ar)}{A_{in}(CO)/A_{in}(Ar)},$$

where $n_{in}(CO)$ and $n_{out}(CO)$ refer to the mole number of CO at the inlet and outlet, respectively, $A_{in}(CO)$ and $A_{in}(Ar)$ refer to the chromatographic peak area of CO and Ar in the feed gas, and $A_{out}(CO)$ and $A_{out}(Ar)$ refer to the chromatographic peak area of CO and Ar in the off-gas.

The selectivity values presented in this work were calculated on a carbon basis.

The selectivity of CO₂ was calculated as

$$S_{CO_2} = \frac{n_{out}(CO_2)}{n_{in}(CO) - n_{out}(CO)} = \frac{f_{CO_2/Ar} [A_{out}(CO_2)/A_{out}(Ar)]}{f_{CO/Ar} [A_{in}(CO)/A_{in}(Ar) - A_{out}(CO)/A_{out}(Ar)]},$$

where $f_{CO_2/Ar}$ is the relative correction factors of CO₂ to Ar, which was determined by the calibrating gas; $A_{out}(CO_2)$ refers to the chromatographic peak area of CO₂ detected by TCD in the off-gas.

Similarly, the selectivity of CH₄ was calculated as

$$S_{CH_4} = \frac{n_{out}(CH_4)}{n_{in}(CO) - n_{out}(CO)} = \frac{f_{CH_4/Ar} [A_{out}(CH_4)/A_{out}(Ar)]}{f_{CO/Ar} [A_{in}(CO)/A_{in}(Ar) - A_{out}(CO)/A_{out}(Ar)]},$$

where $f_{CH_4/Ar}$ is the relative correction factors of CH₄ to Ar, which was determined by the calibrating gas; $A_{out}(CH_4)$ refers to the chromatographic peak area of CH₄ detected by TCD in the off-gas.

The flame ionization detector (FID) were used to detect CH₄ and C₂–C₄

hydrocarbons. The CH₄ selectivity was used as a bridge to calculate the selectivity of C₂–C₄ hydrocarbons identified by FID.

The selectivity for C_xH_y (x = 2–4) hydrocarbons was calculated as

$$S_{C_xH_y} = \frac{x \cdot n_{out}(C_xH_y)}{n_{in}(CO) - n_{out}(CO)} = x \cdot f_{C_xH_y/CH_4} \cdot \frac{A_{FID}(C_xH_y)}{A_{FID}(CH_4)} \cdot S_{CH_4},$$

where $f_{C_xH_y/CH_4}$ is the relative correction factors of C_xH_y to CH₄, which was determined by the calibrating gas; $A_{FID}(CH_4)$ and $A_{FID}(C_xH_y)$ refer to the chromatographic peak area of CH₄ and C_xH_y detected by FID in the off-gas, S_{CH_4} is the CH₄ selectivity calculated by TCD.

Question 10. Any information on the analysis of the C₅+ fraction was reported.

Response:

The liquid and solid products (C₅+) were analyzed offline using an Agilent 7890 gas chromatograph equipped with an HP-5 capillary column connected to a flame ionization detector (FID). The liquid hydrocarbons were dissolved in ethanol, while the solid wax was dissolved in dodecane. The C₅+ products consist of main normal paraffins and a fraction of alkenes. The relative content of each product was detected by the normalization method of peak area. As shown in Figure R13, the carbon number distribution of liquid hydrocarbons mainly concentrates in C₅–C₂₀, while that of solid wax consists a great amount of C₄₀–C₄₆ hydrocarbons.

Figure R13. The carbon number distribution of C₅+ products. (a) Liquid hydrocarbons; (b) Solid wax.

Question 11. Mass balance of around 90% showed a bad analytical practice.

Response:

In the case of Ru/TiO₂ catalytic system, the acquired carbon products for carbon balance calculation consists of CO₂, CH₄, C₂–C₄ and C₅₊ hydrocarbons (including both liquid hydrocarbons and solid wax, which are preserved in cold and hot trap, respectively). The amounts of CO₂ and C₁–C₄ gaseous products can be accurately calculated by GC results, while the amount of C₅₊ fraction was calculated by weighing its mass. Nevertheless, the catalyst filling amount (0.3 g) in the fixed-bed reactor with a space time yield of 0.13 g_{C5+} g_{cat} h⁻¹ makes it difficult to acquire an accurate C₅₊ amount as the hydrocarbon products in cold or hot trap cannot be completely taken out due to the inevitable residue in cold or hot trap. As a result, the carbon balance mainly depends on the amounts of collected C₅₊ products and the amount of C₅₊ fraction calculated by weighing its mass for carbon balance calculation is usually lower than that in reality, which will cause the carbon balance a little lower. As such, the calculated carbon balances greater than 90% are acceptable to apply in product selectivity calculation.

Question 12. *A major challenge of FT is its exothermicity. Support having a good thermal conductivity must be employed, but it is not the case of TiO₂-based materials. Why this choice of Ru/TiO₂ catalytic system instead of other supports which are more potential than TiO₂?*

Response:

We agree with the reviewer that TiO₂ doesn't have a good thermal conductivity to overcome the exothermicity, as compared with Al₂O₃ and SiO₂, which are usually used to support metals in FTS. However, TiO₂ can be used as a promoter in the traditional Ru/Al₂O₃ or Ru/SiO₂ catalysts, as TiO₂ supported Ru catalysts exhibit an enhanced FTS reactivity than Al₂O₃ or SiO₂ supported catalysts. The investigation of Ru/TiO₂ catalysts for FTS can give much guidance to future development of TiO₂ promoted Ru/Al₂O₃ or Ru/SiO₂ catalysts.

Ru is identified to be intrinsic of high activity and selectivity in FTS, and large particle sizes of Ru (~8 nm) are highly desirable [Carballo JMG, et al. *J Catal* **284**,

102–108 (2011); Kang J, Zhang S, Zhang Q, Wang Y. *Angew Chem Int Ed* **48**, 2565–2568 (2009)]^{1,2}, which results in a low utilization of Ru. The utilization of SMSI has been demonstrated to be an alternative strategy to enhance the catalytic reactivity of metal catalysts in FTS [Hernandez Mejia C, van Deelen TW, de Jong KP. *Nat Commun* **9**, 4459–4466 (2018); Kikuchi E, Matsumoto M, Takahashi T, Machino A, Morita Y. *Appl Catal* **10**, 251–260 (1984)]^{17,18}. Besides, the lattice match of RuO₂ and rutile TiO₂ makes great advantage on the size control of Ru NPs on TiO₂. With the small sized Ru, the interface of Ru-TiO₂ can be well engaged into FTS.

In consideration of the moderate thermal conductivity of TiO₂, we use 40 nanometer-level (~40 nm) TiO₂ as support. The Ru/TiO₂ catalyst was further diluted with quartz sand in FTS reaction tests to facilitate the heat transfer.

References

1. Carballo JMG, *et al.* Catalytic effects of ruthenium particle size on the Fischer–Tropsch synthesis. *J Catal* **284**, 102–108 (2011).
2. Kang J, Zhang S, Zhang Q, Wang Y. Ruthenium nanoparticles supported on carbon nanotubes as efficient catalysts for selective conversion of synthesis gas to diesel fuel. *Angew Chem Int Ed* **48**, 2565–2568 (2009).
3. Hwang BJ, *et al.* Probing the formation mechanism and chemical states of carbon-supported Pt-Ru nanoparticles by in situ X-ray absorption spectroscopy. *J Phys Chem B* **110**, 6475–6482 (2006).
4. Shimizu K-i, Oda T, Sakamoto Y, Kamiya Y, Yoshida H, Satsuma A. Quantitative determination of average rhodium oxidation state by a simple XANES analysis. *Appl Catal B* **111–112**, 509–514 (2012).
5. Johnson GR, Werner S, Bell AT. An investigation into the effects of Mn promotion on the activity and selectivity of Co/SiO₂ for Fischer–Tropsch synthesis: evidence for enhanced CO adsorption and dissociation. *ACS Catal* **5**, 5888–5903 (2015).
6. Ciobica IM, van Santen RA. Carbon monoxide dissociation on planar and stepped Ru(0001) surfaces. *J Phys Chem B* **107**, 3808–3812 (2003).
7. Reuter K, Scheffler M. Composition, structure, and stability of RuO₂(110) as a function of oxygen pressure. *Phys Rev B* **65**, 035406 (2001).
8. Lide DR. CRC Handbook of Chemistry and Physics, 88th Edition. *Taylor & Francis Group, Boca Raton*, (2007).
9. Lejaeghere K, *et al.* Reproducibility in density functional theory calculations of solids. *Science* **351**, aad3000 (2016).
10. McEvoy AJ. ESCA spectrum and band structure of ruthenium chloride. *phys stat sol (a)* **71**, 569–574 (1982).
11. Pollini II. Photoemission study of the electronic structure of CrCl₃ and RuCl₃ compounds. *Phys Rev B* **50**, 2095–2103 (1994).
12. Morgan DJ. Resolving ruthenium: XPS studies of common ruthenium

- materials. *Surf Interface Anal* **47**, 1072–1079 (2015).
13. Tauster SJ, Fung SC, Garten RL. Strong metal-support interactions. Group 8 noble metals supported on titanium dioxide. *J Am Chem Soc* **100**, 170–175 (1978).
 14. Tauster SJ, Fung SC, Baker RTK, Horsley JA. Strong interactions in supported-metal catalysts. *Science* **211**, 1121–1125 (1981).
 15. Tauster SJ. Strong metal–support interactions. *Acc Chem Res* **20**, 389–394 (1987).
 16. Green CL, Kucernak A. Determination of the platinum and ruthenium surface areas in platinum-ruthenium alloy electrocatalysts by underpotential deposition of copper. I. unsupported catalysts. *J Phys Chem B* **106**, 1036–1047 (2002).
 17. Hernandez Mejia C, van Deelen TW, de Jong KP. Activity enhancement of cobalt catalysts by tuning metal-support interactions. *Nat Commun* **9**, 4459–4466 (2018).
 18. Kikuchi E, Matsumoto M, Takahashi T, Machino A, Morita Y. Fischer-Tropsch synthesis over titania-supported ruthenium catalysts. *Appl Catal* **10**, 251–260 (1984).

REVIEWERS' COMMENTS:

Reviewer #2 (Remarks to the Author):

Rebuttal and actions suffice in my opinion. As before, I advise positively about publication of the paper in Nature Communications.

Reviewer #3 (Remarks to the Author):

Manuscript title: Tuning reactivity of Fischer–Tropsch synthesis by regulating TiO_x overlayer over Ru/TiO₂ nanocatalysts

Manuscript reference: NCOMMS-19-39843A

This work reports the catalytic performance of Ru/TiO₂ catalysts, pretreated at different reduction temperatures, in FT synthesis. It particularly evidenced the contribution of TiO_x layers formed by reduction at high temperature in the activation of CO in Fisher-Tropsch synthesis.

Some small details would be improved:

- Keywords: Add “CO activation”

- A list of abbreviations can be useful (if applicable)

The revised version satisfactorily addressed all the questions and comments of the reviewers. I propose to accept the publication of this paper in Nature Communications.

Point-by-point response to the referees

REVIEWERS' COMMENTS:

Reviewer #2 (Remarks to the Author):

Rebuttal and actions suffice in my opinion. As before, I advise positively about publication of the paper in Nature Communications.

Response: Thank you again for your valuable comments and suggestions.

Reviewer #3 (Remarks to the Author):

Manuscript title: Tuning reactivity of Fischer–Tropsch synthesis by regulating TiO_x overlayer over Ru/TiO₂ nanocatalysts

Manuscript reference: NCOMMS-19-39843A

This work reports the catalytic performance of Ru/TiO₂ catalysts, pretreated at different reduction temperatures, in FT synthesis. It particularly evidenced the contribution of TiO_x layers formed by reduction at high temperature in the activation of CO in Fisher-Tropsch synthesis.

Some small details would be improved:

- Keywords: Add “CO activation”*
- A list of abbreviations can be useful (if applicable)*

The revised version satisfactorily addressed all the questions and comments of the reviewers. I propose to accept the publication of this paper in Nature Communications.

Response: Thank you for your kind suggestion. We have added "CO activation" in the Keywords. As each new abbreviation in this paper have been clearly defined at its first appearance, a list of abbreviations is not included in the final version of this paper.